# Enhancing Cost Efficiency in Active Learning with Candidate Set Query

**Yeho Gwon**[*]                                                                                          *yeho.gwon@postech.ac.kr*
*Department of Computer Science and Engineering*
*POSTECH*

**Sehyun Hwang**[*]                                                                                        *sehyun03@postech.ac.kr*
*Department of Computer Science and Engineering*
*POSTECH*

**Hoyoung Kim**                                                                                           *cskhy16@postech.ac.kr*
*Graduate School of Artificial Intelligence*
*POSTECH*

**Jungseul Ok**                                                                                           *jungseul@postech.ac.kr*
*Graduate School of Artificial Intelligence*
*POSTECH*

**Suha Kwak**                                                                                             *suha.kwak@postech.ac.kr*
*Graduate School of Artificial Intelligence*
*POSTECH*

**Reviewed on OpenReview:** *https://openreview.net/forum?id=LhHxl30xQ1*

## Abstract

This paper introduces a cost-efficient active learning (AL) framework for classification, featuring a novel query design called *candidate set query*. Unlike traditional AL queries requiring the oracle to examine all possible classes, our method narrows down the set of candidate classes likely to include the ground-truth class, significantly reducing the search space and labeling cost. Moreover, we leverage conformal prediction to dynamically generate small yet reliable candidate sets, adapting to model enhancement over successive AL rounds. To this end, we introduce an acquisition function designed to prioritize data points that offer high information gain at lower cost. Empirical evaluations on CIFAR-10, CIFAR-100, and ImageNet64x64 demonstrate the effectiveness and scalability of our framework. Notably, it reduces labeling cost by 48% on ImageNet64x64. The project page can be found at https://yehogwon.github.io/csq-al.

## 1 Introduction

Deep neural networks owe much of their success to large-scale annotated datasets (Deng et al., 2009; Kirillov et al., 2023; OpenAI, 2023; Radford et al., 2021). Scaling datasets is crucial for improving both of their performance (Hestness et al., 2017; Zhai et al., 2022) and robustness (Fang et al., 2022). However, the resources demanded for manual annotation pose a significant bottleneck, particularly in fields requiring expert input like medical data. In response to these challenges, cost-efficient methods for dataset collection, such as semi-automatic labeling (Kim et al., 2024; Qu et al., 2024; Wang et al., 2024), synthetic data generation (Liu et al., 2019a; Tran et al., 2019), and active learning (AL) (Ash et al., 2020; Kirsch et al., 2019; Sener & Savarese, 2018; Settles, 2009; Sinha et al., 2019; Wang & Ye, 2015) have been studied.

---

[*]equal contribution.

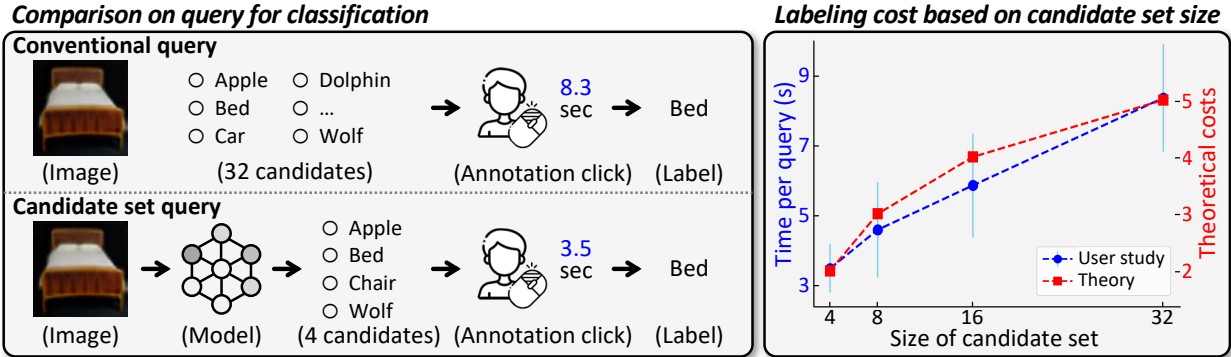

Figure 1: Conventional query versus candidate set query. (*left*) While the conventional query presents all possible options to annotators, CSQ leverages the knowledge of the model to offer narrowed options that are likely to include the ground-truth label, thereby reducing the annotation time. (*right*) By conducting a user study on 40 participants, we demonstrate that the labeling cost increases logarithmically to the candidate set size, which closely aligns with the information-theoretic cost suggested by Hu et al. (2020) with a correlation coefficient of 0.97. Note that as the labeling cost increases per sample, the overall labeling cost increases significantly when multiplied by the total number of labeled samples. Further details of the user study are provided in Sec. 4.2 and Appendix A.

This paper investigates AL for classification, where a training algorithm selects informative samples from the data pool and queries annotators for their class labels within a limited budget. We focus on improving the design of annotation queries, emphasizing their critical role. To be specific, we consider image classification of $L$ classes. In the conventional query design, an annotator is asked to choose a class from a list of $L$ classes. Here, the effort needed to review the entire class list and identify the correct class increases as the list size $L$ increases; according to an information-theoretic analysis (Hu et al., 2020), the cost of choosing among $L$ options is $\log_2 L$. To address this issue of growing annotation cost, recent studies (Hu et al., 2020; Kim et al., 2024) employ a 1-bit query design asking annotators to check if the top-1 model prediction is correct. While this simplifies and speeds up annotation, it produces weak supervision incompatible with standard classification loss functions, necessitating specialized losses and algorithms like contrastive loss and semi-supervised learning techniques.

We propose *candidate set query* (CSQ), a novel AL query design that remains cost-efficient with increasing classes and integrates seamlessly with existing loss functions. CSQ presents the annotator with an image and a narrowed set of candidate classes, which is likely to include the ground-truth class. The annotator first searches this small candidate set for the ground-truth class and proceeds to the remaining classes *only if* the ground-truth class is not found during the first search. This query approach can reduce labeling cost by reducing the search space required for annotation, which is particularly effective in scenarios with a wide range of classes where the search space for the annotator could be extensive. Figure 1(*left*) compares CSQ with the conventional query in AL for classification to show its efficiency.

In the CSQ framework, the design of the candidate set is crucial for its effectiveness. On one hand, too many candidates unnecessarily increase the labeling costs. On the other hand, too few candidates are likely to omit the ground-truth class, requiring an additional query to identify the ground-truth class among the remaining classes, which is more expensive than the conventional query. To enhance the effectiveness of the CSQ framework, we propose to construct candidate sets guided by prediction uncertainty from a trained model using conformal prediction (Shafer & Vovk, 2008; Angelopoulos et al., 2023). Conformal prediction aims at constructing a set of predictions including the true class, where each set is properly sized based on the certainty of the model about the input. This strategy enables flexible adjustment of the candidate set for each sample, expanding it for an uncertain sample to include the true label and shrinking it for a more certain one to reduce the labeling cost. Furthermore, we optimize the level of certainty in conformal prediction to minimize the labeling cost for each round. Therefore, this candidate set construction adapts

to the increasing accuracy of the model over successive AL rounds, refining the candidate set as the model improves.

Last but not least, we propose a new acquisition function designed to maximize the cost efficiency of CSQ. Conventional acquisition functions in AL are designed to favor samples with high estimated information gain, assuming uniform annotation costs across all samples. On the other hand, in CSQ, the labeling cost for each sample varies according to the size of its candidate set. Thus, we propose an acquisition function that evaluates samples based on the ratio of estimated information gain to labeling cost. Specifically, we combine the conventional acquisition function score, which indicates the estimated information gain, with the estimated cost derived from the candidate set, favoring samples that maximize information gain per unit cost. This cost-efficient acquisition function can incorporate with any sample-wise acquisition score, ensuring the selection of both informative and cost-efficient samples.

The proposed method achieves state-of-the-art performance on CIFAR-10 (Krizhevsky et al., 2009), CIFAR-100 (Krizhevsky et al., 2009), and ImageNet64x64 (Chrabaszcz et al., 2017). We verify the effectiveness and generalizability of CSQ through extensive experiments with varying datasets, acquisition functions, and budgets. Notably, CSQ achieves the same performance as the conventional query on ImageNet64x64 at only 48% of the cost, showing its scalability. Our ablation studies demonstrate that both our candidate set construction and sampling strategy contribute to the performance. Furthermore, the necessity of CSQ is demonstrated by a user study involving 40 participants. In short, the main contribution of this paper is four-fold:

- We propose a novel query design for active learning, where the annotator is presented with an image and a narrowed set of candidate classes that is likely to include the ground-truth class. This approach, termed CSQ, reduces labeling cost by minimizing the search space the annotator needs to explore.

- To maximize the advantage of CSQ, we propose to utilize conformal prediction to dynamically generate small yet reliable candidate sets optimized to reduce labeling costs, adapting to the evolving model throughout successive AL rounds.

- We propose a new acquisition function that prioritizes a data point expected to have high information gain relative to its labeling cost, enhancing cost efficiency.

- The proposed framework achieved state-of-the-art performance on diverse image classification datasets, CIFAR-10, CIFAR-100, and ImageNet64x64, showing its effectiveness and generalizability.

## 2 Related Work

**Acquisition functions in AL.** The key to AL is to select and annotate the most informative samples (Settles, 2009; Dasgupta, 2011; Hanneke et al., 2014). To assess informativeness, various acquisition functions have been proposed, considering either the uncertainty of model predictions (Asghar et al., 2017; He et al., 2019; Ostapuk et al., 2019; Fuchsgruber et al., 2024; Kim et al., 2024; Cho et al., 2024; Kim et al., 2023; Wang et al., 2022), diversity in feature space (Sener & Savarese, 2018; Sinha et al., 2019; Yehuda et al., 2022), or both (Ash et al., 2020; Hwang et al., 2022; Wang & Ye, 2015; Wang et al., 2019; Hacohen et al., 2022; Hacohen & Weinshall, 2023). Disagreement-based AL and its variants are supported by rigorous theoretical learning guarantees (Hanneke et al., 2014; Krishnamurthy et al., 2019). However, these methods assume uniform sample costs and select based solely on the amount of information. We emphasize that the labeling cost required for each sample varies and prioritize samples offering the best information-to-cost ratio.

**Efficient query design.** Designing efficient annotation queries reduces the annotation costs of crafting datasets. In AL, diverse types of queries have been investigated, including conventional classification queries (Huang et al., 2015; Kang et al., 2020; Yu et al., 2020; Xie & Huang, 2021; Cour et al., 2011), one-bit queries (Hu et al., 2020; Joshi et al., 2010) asking for yes or no answers, multi-class queries (Hwang et al., 2023) identifying all classes within a set of multiple instances, relative queries (Qian et al., 2013) asking for similarity of triplets, and correction queries (Kim et al., 2024) utilizing pseudo labels from the model. While these query methods require tailored loss functions, our candidate set query (CSQ) is cost-efficient

and provides complete supervision, integrating seamlessly with existing loss functions. The approach closely related to CSQ is the $n$-ary query (Bhattacharya & Chakraborty, 2019), which reduces the search space by asking for the correct class among top-$n$ predictions of the model. However, the $n$-ary query uses a fixed number of top-$n$ predictions for all data without considering individual sample difficulty. On the other hand, CSQ adjusts the candidate set size based on sample difficulty and model performance using conformal prediction. Through rigorous comparisons, we demonstrate that CSQ achieves a superior model performance at the same cost compared to the previous query designs.

**Realistic cost model for AL.** We consider annotation cost, rather than the number of labeled samples, as the primary objective of active learning. While we follow Hu et al. (2020) in modeling annotation cost based on the size of the labeling options using an information-theoretic perspective, other works have explored additional factors influencing real-world annotation cost, such as annotator behavior and interaction complexity (Settles et al., 2008; Arora et al., 2009; Wallace et al., 2011; Herde et al., 2021). For example, Settles et al. (2008) and Arora et al. (2009) show that instance, task, and user features jointly affect annotation time, while Wallace et al. (2011) propose a cost-saving strategy where less experienced annotators can pass uncertain cases to more experienced ones. Herde et al. (2021) provide a taxonomy of cost-aware active learning approaches under more realistic assumptions. These studies highlight that annotation cost depends on multiple factors and varies across settings. While our size-based proxy is effective and supported by user study, richer cost models may better capture real-world annotation behavior and further improve performance.

**Conformal prediction (CP).** CP enables us to quantify uncertainty in predictions with an associated confidence level (Shafer & Vovk, 2008). Recent advances in CP empower classifiers to generate predictive sets that include the ground-truth label with a probability chosen by the user (Angelopoulos et al., 2020; 2023). In the field of AL, nonconformity measurements from CP are employed in the acquisition function to select informative samples (Matiz & Barner, 2020). In contrast, we utilize CP not only to develop a cost-efficient acquisition function but also to design an efficient candidate set query, reducing the labeling cost.

## 3 Proposed Method

We consider general classification tasks such that for input $\mathbf{x}$ and a categorical variable $y \in \mathcal{Y} = \{1, 2, \ldots, L\}$, a model parameterized by $\boldsymbol{\theta}$ predicts the class of the input as $\arg\max_{y \in \mathcal{Y}} P_{\boldsymbol{\theta}}(y|\mathbf{x})$. We study an active learning (AL) scenario conducted over $R$ rounds. In each round $r$, a budget of $B$ samples with high acquisition function values is actively selected from the unlabeled data pool $\mathcal{X}$. This actively selected set $\mathcal{A}_r$ is then labeled by an annotator to form the labeled dataset $\mathcal{D}_r$ with labeling cost $C_r$, and is used to update the model. Let $\boldsymbol{\theta}_r$ denote the model trained on the accumulated labeled data up to round $r$, $\bigcup_{i=0}^{r} \mathcal{D}_i$. Our goal is to maximize the performance of $\boldsymbol{\theta}_r$, while minimizing the accumulated cost $\bigcup_{i=0}^{r} C_i$. The key aspect of the proposed method is candidate set query (CSQ), which reduces $C_r$ by narrowing the set of candidate classes presented to annotators. For simplicity, we omit the round index $r$ from $\boldsymbol{\theta}_r$ in the remainder of this section.

In the following, we first introduce CSQ and discuss its efficiency in labeling cost (Sec. 3.1). Then, we present a method to construct a candidate set based on the prediction uncertainty of a trained model for a given sample (Sec. 3.2). Lastly, we introduce an acquisition function designed to consider cost efficiency as well as information gain (Sec. 3.3). The overall pipeline of the CSQ framework combined with our cost-efficient sampling is summarized in Algorithm 1.

### 3.1 Candidate set query

CSQ for an instance $\mathbf{x}$ is associated with a (non-empty) candidate set $Y(\mathbf{x}) \subseteq \mathcal{Y}$ such that $1 \leq |Y(\mathbf{x})| \leq L$. CSQ first asks the annotator to choose the ground-truth class in $Y(\mathbf{x})$ if it exists, or to verify the absence of the ground-truth label in $Y(\mathbf{x})$, *i.e.*, the annotator is first asked to pick an option out of $(k+1)$ choices, where $k = |Y(\mathbf{x})|$. Only if the absence of the ground-truth class in the candidate set is verified, the annotator is further asked to select the ground-truth class from the remaining ones $\mathcal{Y} \setminus Y(\mathbf{x})$. In short, CSQ asks the following query with an image to the oracle.

---
**Algorithm 1** Cost-efficient active learning with candidate set query
---
**Require:** The number of AL rounds $R$, per-round budget $B$, unlabeled data pool $\mathcal{X}$, initial labeled dataset $\mathcal{D}_0$.
 1: Train the initial model $\boldsymbol{\theta}_0$ on $\mathcal{D}_0$.
 2: **for** $r = 1, 2, \ldots, R$ **do**
 3:     Estimate sample-wise labeling cost of $\mathbf{x} \in \mathcal{X}$.
 4:     Select $B$ samples $\mathcal{A}_r \subset \mathcal{X}$ considering both their estimated labeling cost and informativeness.     ▷ Sec. 3.3
 5:     Construct candidate set $\hat{Y}(\mathbf{x})$ for each $\mathbf{x} \in \mathcal{X}$.     ▷ Sec. 3.2
 6:     Query annotator for label $y$ of $\mathbf{x} \in \mathcal{A}_r$ using candidate set $\hat{Y}(\mathbf{x})$ to form $\mathcal{D}_r$.
 7:     Get model $\boldsymbol{\theta}_r$ trained on $\bigcup_{i=0}^{r} \mathcal{D}_i$.
 8: **end for**
 9: **Return** Final model $\boldsymbol{\theta}_R$.

---

*Select the ground-truth class from the candidate set, or choose* "None of the above."
*(only if* "None of the above" *is chosen): Select the ground-truth from the remaining classes.*

Following the information-theoretic cost model of (Hu et al., 2020) and the user-study results in Table 1, we model the cost of selecting one label from $k$ candidates as $\log_2 k$. Then, the labeling cost $\Gamma(Y(\mathbf{x}), y)$ of CSQ for input $\mathbf{x}$, ground-truth label $y$, and candidate set $Y(\mathbf{x}) \subseteq \mathcal{Y}$ can be obtained as:

$$\Gamma(Y(\mathbf{x}), y) = \begin{cases} \log_2(k+1) & \text{if } y \in Y(\mathbf{x}) \\ \log_2(k+1) + \log_2(L-k) & \text{otherwise} \end{cases}. \tag{1}$$

The conventional query in AL is a special case of CSQ where $Y(\mathbf{x}) = \mathcal{Y}$, and it is inefficient since the annotator must search through the entire set of size $L$ with a cost of $\log_2 L$. The following theorem reveals the condition under which the expected cost of CSQ offers an improvement over that of the conventional query.

**Theorem 3.1.** *Assume the information-theoretic cost model (Hu et al., 2020) of selecting one out of $L$ possible options to be $\log_2 L$. Let $L \geq 2$ be the number of classes, $k = |Y(\mathbf{x})|$, and $\alpha$ be the probability that the candidate set $Y(\mathbf{x})$ does not include the ground-truth class of instance $\mathbf{x}$. Denote by $C_{\mathrm{con}}$ and $C_{\mathrm{csq}}$ the expected costs of the conventional query and the candidate set query, respectively. If*

$$\frac{\log_2(k+1)}{\log_2 L} < 1 - \alpha, \tag{2}$$

*then the candidate set query is strictly cheaper, i.e., $C_{\mathrm{csq}}(L, \mathbf{x}, \alpha) < C_{\mathrm{con}}(L, \mathbf{x})$.*

*Proof.* Recalling the definition of $\alpha$, we have $C_{\mathrm{csq}}(L, \mathbf{x}, \alpha) = (1-\alpha)\log_2(k+1) + \alpha\{\log_2(k+1) + \log_2(L-k)\}$ from Eq. (1). As $L - k < L$, the cost ratio of $C_{\mathrm{csq}}(L, \mathbf{x}, \alpha)$ to $C_{\mathrm{con}}(L, \mathbf{x})$ for instance $\mathbf{x}$ is induced as:

$$\frac{C_{\mathrm{csq}}(L, \mathbf{x}, \alpha)}{C_{\mathrm{con}}(L, \mathbf{x})} = \frac{\log_2(k+1) + \alpha\log_2(L-k)}{\log_2 L}$$
$$< \frac{\log_2(k+1)}{\log_2 L} + \alpha. \tag{3}$$

Although we adopt the cost model from Hu et al. (2020), Theorem 3.1 holds for any cost model that increases monotonically with the number of options.

**Remark 3.2.** *If we constrain all candidate set sizes $k$ to be fixed, then $1 - \alpha$ corresponds to the top-$k$ accuracy $p_k$ of the model. Therefore, when $p_k \geq \log_L(k+1)$, CSQ consistently offers an improvement over the conventional query in the expected labeling cost. For example, in datasets such as CIFAR-10 ($L = 10$), CIFAR-100 ($L = 100$), and ImageNet ($L = 1000$), if the model has a top-1 accuracy (i.e., $k = 1$) of at least 30.1%, 15.1%, and 10.0% respectively, then CSQ always provides an improvement.*

The above proof and remark demonstrate that under moderate conditions, CSQ is more efficient than the conventional query. As described in Eq. (3), the cost of CSQ decreases as both $\alpha$ and $k$ become smaller. However, since $k$ and $\alpha$ are inversely related, balancing the trade-off between $\alpha$ and $k$ is essential to fully leverage CSQ. Also, fixing candidate set sizes as in Remark 3.2 is suboptimal because it does not consider the difficulty of individual samples. In the following section, we introduce our candidate set construction method, which both reflects the uncertainty of each sample and automatically balances the trade-off between $\alpha$ and $k$.

### 3.2 Construction of cost-efficient candidate set

As shown in Eq. (1) and Theorem 3.1, a candidate set needs to be both small and accurate in covering the ground-truth class. To do so, we propose using conformal prediction (Romano et al., 2020) to get a reliable and cost-optimized candidate set using the trained model $\boldsymbol{\theta}$ of the previous round.

**Calibration set collection.** Conformal prediction requires a labeled set for calibration that has not been used during the model training phase; this set must follow the same distribution as the target dataset for prediction (Vovk et al., 1999; Angelopoulos et al., 2023). To achieve this, we randomly select $n_{\text{cal}}$ samples from the actively selected data $\mathcal{A}_r$ and annotate them within the given budget to form $\mathcal{D}_{\text{cal}} = \{(\mathbf{x}_i, y_i)\}_{i=1}^{n_{\text{cal}}}$. The calibration set $\mathcal{D}_{\text{cal}}$ is used for conformal prediction and candidate set optimization, which will be explained in the following sections. Note that $\mathcal{D}_{\text{cal}}$ also contributes to model training after the candidate set construction.

**Conformal prediction.** Using the model $\boldsymbol{\theta}$ from the previous round and calibration set $\mathcal{D}_{\text{cal}} \subset \mathcal{A}_r$ of size $n_{\text{cal}}$, we define a collection of conformal scores $\mathbf{s} := (s_i)_{i=1}^{n_{\text{cal}}}$, where $s_i := 1 - P_{\boldsymbol{\theta}}(y_i \mid \mathbf{x}_i)$ for $(\mathbf{x}_i, y_i) \in \mathcal{D}_{\text{cal}}$. Then, we obtain the $(1 - \alpha)$ empirical quantile $\hat{Q}(\alpha)$ of $\mathbf{s}$, indicating that at least $100 \times (1 - \alpha)\%$ of the scores in $\mathbf{s}$ are smaller than $\hat{Q}(\alpha)$. This empirical quantile $\hat{Q}(\alpha)$ is given as,

$$\hat{Q}(\alpha) := \min_{s \in \mathbf{s}} \left\{ s : \frac{1}{n_{\text{cal}}} \sum_{s' \in \mathbf{s}} \left( \mathbb{1}[s' \leq s] \right) \geq 1 - \alpha \right\} , \tag{4}$$

where $\alpha \in (0, 1)$ is an error rate hyperparameter, and $\mathbb{1}[\cdot]$ is an indicator function. Then, we define the candidate set $\hat{Y}_{\boldsymbol{\theta}}(\mathbf{x}, \alpha)$ for unlabeled data $\mathbf{x}$ using conformal prediction as follows:

$$\hat{Y}_{\boldsymbol{\theta}}(\mathbf{x}, \alpha) := \left\{ y : P_{\boldsymbol{\theta}}(y | \mathbf{x}) \geq 1 - \hat{Q}(\alpha),\ y \in \mathcal{Y} \right\} . \tag{5}$$

Previous study (Vovk et al., 1999; Angelopoulos et al., 2023) proved that the candidate set includes the true label with the probability not less than $1 - \alpha$, which is,

$$P\big(y \in \hat{Y}_{\boldsymbol{\theta}}(\mathbf{x}, \alpha)\big) \geq 1 - \alpha . \tag{6}$$

This ensures the inclusion of the ground-truth classes even under model overconfidence, while adaptively reflecting uncertainties throughout the AL process. Without loss of generality, we consider $\hat{Y}_{\boldsymbol{\theta}}(\mathbf{x}, 0) = \mathcal{Y}$, where $\mathcal{Y}$ corresponds to the conventional query. More detailed procedure of conformal prediction is in Appendix C.

**Cost-optimized error rate selection.** The proposed candidate set construction method (Eq. (5)) adapts the size of the candidate set for each sample based on its predicted uncertainty. While this allows the candidate set to be both compact and reliable, it requires manually adjusting the hyperparameter $\alpha$. To eliminate the need for manual tuning, we introduce an automatic selection scheme that optimizes $\alpha$ at each AL round by minimizing the expected labeling cost on the calibration set, which serves as a pseudo-validation set. To be specific, $\alpha$ is optimized by

$$\alpha^* := \underset{\alpha \in [0,1)}{\arg\min} \sum_{(\mathbf{x}, y) \in \mathcal{D}_{\text{cal}}} \Gamma(\hat{Y}_{\boldsymbol{\theta}}(\mathbf{x}, \alpha), y) , \tag{7}$$

where $\Gamma(\cdot, y)$ is the labeling cost in Eq. (1). We implement this optimization using a grid search over predefined options of $\alpha$. The optimization is computationally efficient, requiring only negligible computation

as shown in Figure 7. This approach not only eliminates the reliance on hand-tuned hyperparameters but also helps construct candidate sets in a more cost-efficient manner, as the selected $\alpha^*$ is tailored to minimize the labeling cost under the proposed cost model for each round. Since the optimization in Eq. (7) naturally considers the conventional query as a special case of CSQ at $\alpha = 0$, CSQ is at least as efficient as, and in general more efficient than, the conventional query.

Note that to construct the candidate set query, the calibration set $\mathcal{D}_{\text{cal}}$ is required to calculate $(1 - \alpha^*)$ quantile in Eq. (4). Thus, when getting annotations of $\mathcal{D}_{\text{cal}}$ in the calibration set collection step, the candidate set query of the current round cannot be applied. To avoid this circular dependency, the quantile from the previous round is used when labeling $\mathcal{D}_{\text{cal}}$.

### 3.3 Cost-efficient acquisition function

Since the labeling cost of each sample varies in CSQ, we propose to consider the cost for annotation in the acquisition function. We implement an acquisition function that evaluates samples based on the ratio of the estimated information gain to the estimated labeling cost. The information gain is quantified using an established acquisition score, entropy, BADGE (Ash et al., 2020), ProbCover (Yehuda et al., 2022), or SAAL (Kim et al., 2023), though our approach is compatible with other acquisition scores as well, not just these. Given a conventional acquisition score $g_{\text{score}}(\mathbf{x})$, the proposed cost-efficient acquisition function $g_{\text{cost}}$ is given by,

$$g_{\text{cost}}(\mathbf{x}) := \frac{(1 + g_{\text{score}}(\mathbf{x}))^d}{\log_2(k+1) + \alpha^* \log_2(L-k)} \ , \tag{8}$$

where $d$ is a hyperparameter adjusting the influence of $g_{\text{score}}(\mathbf{x})$ and $\alpha^*$ is the optimized error rate hyperparameter obtained by Eq. (7). The denominator is an expected cost derived from our cost model (Eq. (1)), considering two cases: the correct label is included or excluded from the candidate set, which is $(1 - \alpha^*)\log_2(k+1) + \alpha^*\{\log_2(k+1) + \log_2(L-k)\}$. This expected cost assumes the candidate set to include the ground-truth class with a probability of $1 - \alpha^*$, which is supported by the coverage guarantee in Eq. (6).

## 4 Experiments

### 4.1 Experimental setup

**Datasets.** We use three image classification datasets: CIFAR-10 (Krizhevsky et al., 2009), CIFAR-100 (Krizhevsky et al., 2009), and ImageNet64x64 (Chrabaszcz et al., 2017). CIFAR-10 comprises 50K training and 10K validation images across 10 classes. CIFAR-100 contains the same number of images as CIFAR-10, but is associated with 100 classes. ImageNet64x64 is a downsampled version of ImageNet (Deng et al., 2009) with a resolution of $64 \times 64$, which consists of 1.2M training and 50K validation images with 1000 classes. Following previous studies, we evaluate a model using the validation split of each dataset.

**Implementation details.** For CIFAR-10 and CIFAR-100, we adopt ResNet18 (He et al., 2016) as a classification model. We train it for 200 epochs using AdamW (Loshchilov & Hutter, 2019) optimizer with an initial learning rate of $1e{-}3$, decreasing by a factor of 0.2 at epochs 60, 120, and 160. We apply a weight decay of $5e{-}4$ and a data augmentation consisting of random crop, random horizontal flip, and random rotation. For ImageNet64x64, we adopt WRN-36-5 (Zagoruyko, 2016), and train it for 30 epochs using AdamW optimizer with an initial learning rate of $8e{-}3$. We apply a learning rate warm-up for 10 epochs from $2e{-}3$. After the warm-up, we decay the learning rate by a factor of 0.2 every 10 epochs. We adopt random horizontal flip and random translation as data augmentation. For all the datasets, we use Mix-up (Zhang et al., 2018), where a mixing ratio is sampled from $\text{Beta}(1, 1)$. We set the size of the calibration dataset $n_{\text{cal}}$ to 500 for CIFAR-10 and CIFAR-100, and 5K for ImageNet64x64. For all datasets and acquisition functions, hyperparameter $d$ in Eq. (8) is set to 0.3.

**Active learning protocol.** For CIFAR-10, we conduct 10 AL rounds of consecutive data sampling and model updates, while for CIFAR-100, we perform 9 AL rounds. In both cases, the per-round budget is 6K images. For ImageNet64x64, we conduct 16 AL rounds with a per-round budget of 60K images. The detailed

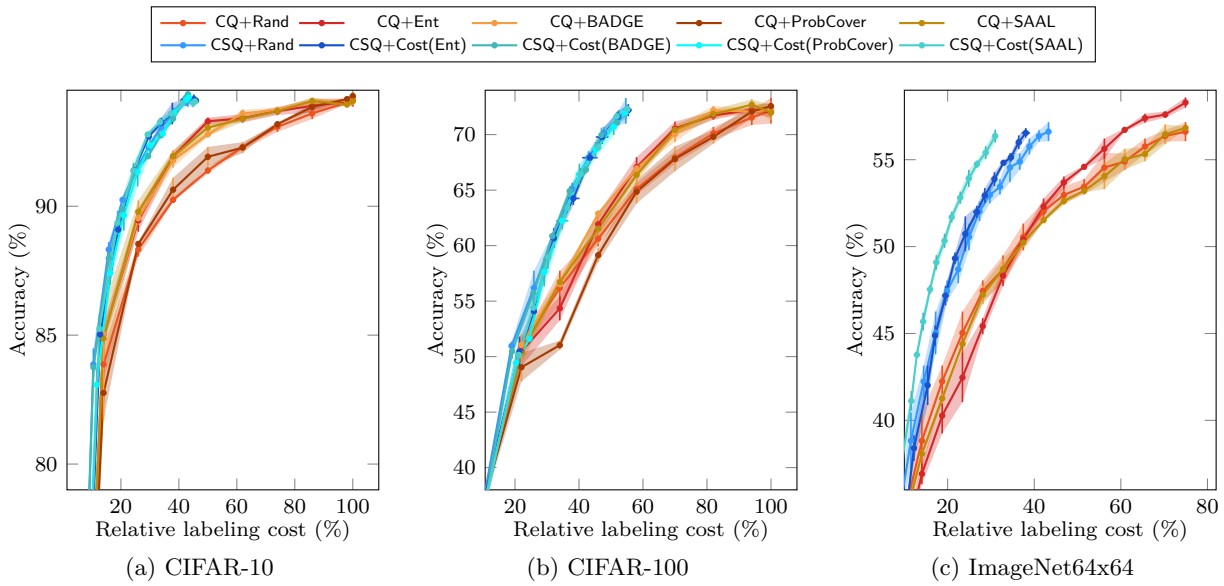

Figure 2: Accuracy (%) versus relative labeling cost (%) for conventional query (CQ) and the proposed candidate set query (CSQ) with different acquisition functions. CSQ approaches (blue lines) consistently outperform the CQ baselines (red lines) by a significant margin across various cost budgets, acquisition functions, and datasets[1].

budget configuration for the three datasets is shown in Table 5. In the initial round, we randomly sample 1K images for CIFAR-10, 5K images for CIFAR-100, and 60K images for ImageNet64x64. In each round, the model is evaluated based on two factors: its accuracy (%) on the validation set, and the accumulated annotation cost required to train it. The annotation cost is defined as a relative labeling cost (%) compared to the cost of labeling the entire training set using the conventional query, given by $N \log_2 L$, where $N$ is the size of the entire training set, and $L$ is the number of classes. We conduct all experiments with three independent trials with different random seeds and report the mean and standard deviation to ensure reproducibility.

**Baseline methods.** We compare our candidate set query (CSQ) with the conventional query (CQ) in combination with various sampling strategies. To be specific, we employ random (Rand), entropy (Ent), BADGE (Ash et al., 2020), ProbCover (Yehuda et al., 2022), and SAAL (Kim et al., 2023) as the sampling strategies. Cost(·) indicates the proposed cost-efficient sampling (Eq. (8)) using conventional acquisition scores; *e.g.*, Cost(SAAL) is the one combined with SAAL. We denote the combination of the query and sampling method with '+', *e.g.*, CSQ+Rand is a candidate set query combined with random sampling.

## 4.2 Experimental results

**Candidate set query vs. Conventional query.** In Figure 2, we compare the performance of the candidate set query (CSQ) with the conventional query (CQ) on CIFAR-10, CIFAR-100, and ImageNet64x64 with different acquisition functions. CSQ approaches consistently outperform the CQ approaches across various acquisition functions and datasets, demonstrating the general effectiveness of our method. Notably, CSQ reduces the labeling cost by 43%, 54%, and 48% on CIFAR-10, CIFAR-100, and ImageNet64x64, compared to CQ, respectively. This is promising as it shows that the same volume of labeled data can be obtained at roughly half the cost, without introducing any label noise or sample bias. Notably, the performance gain of CSQ increases as the model improves, as it is tailored to the improved model. In the appendix, we also present experiments on a text classification task (Figure 14) showing the generalization ability of the proposed method to the natural language domain. Additionally, we provide the zoomed version of Figure 2 in Figure 19 and Figure 20.

---

[1]Unfortunately, for ImageNet64x64, we exclude the BADGE and ProbCover acquisition baselines due to their computational intractability.

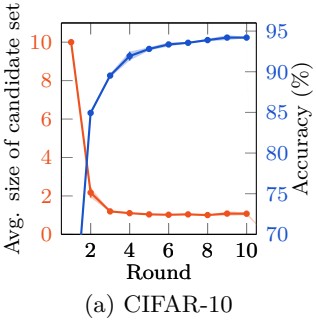
(a) CIFAR-10

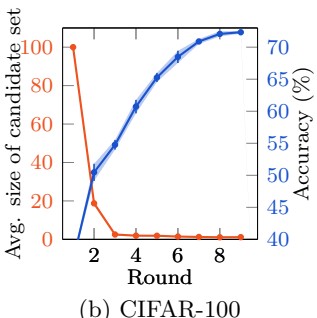
(b) CIFAR-100

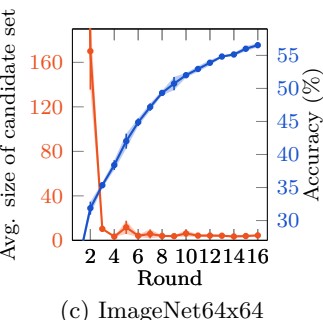
(c) ImageNet64x64

Figure 3: Average size of the candidate set and accuracy (%) of our method with cost-efficient entropy sampling in varying rounds on CIFAR-10, CIFAR-100, and ImageNet64x64. Note that the candidate set includes all classes at the initial round, but our candidate set design effectively reduces its size over successive active learning rounds, adapting dynamically as the model accuracy improves.

Table 1: User-study results on a fixed image set with different numbers of class options presented to the annotator. A smaller option set yields both faster annotation and higher accuracy. The empirical trends also align closely with the theoretical cost curves in Figure 1(*right*). In all experiments, we treat annotation cost as proportional to annotation time, which we consider a more practical measure than simply counting labeled examples.

| Class options (#) | 4 | 8 | 16 | 32 |
|---|---|---|---|---|
| Annotation Time (s) | $\mathbf{69.4}_{\pm 13.8}$ | $91.5_{\pm 27.3}$ | $116.9_{\pm 29.6}$ | $166.9_{\pm 30.8}$ |
| Accuracy (%) | $\mathbf{100.0}_{\pm 0.0}$ | $98.5_{\pm 3.2}$ | $99.5_{\pm 1.5}$ | $95.5_{\pm 5.2}$ |

**Progressive reduction in candidate set size.** The effectiveness of CSQ stems from its ability to reduce labeling costs through smaller candidate sets. To demonstrate this, Figure 3 shows the average size of the candidate sets and accuracy (%) of our method with varying AL rounds on CIFAR-10, CIFAR-100, and ImageNet64x64. After the first round, CSQ achieves a sufficiently small candidate set size and continues to reduce it as accuracy improves, thereby enhancing labeling efficiency.

**Empirical validation for our cost model.** We conduct a user study with 40 annotators who label samples using candidate sets of various sizes; see Appendix A for more details. The results in Table 1 suggest that shrinking candidate sets improves both labeling efficiency and accuracy. They also align closely with the theoretical cost (Hu et al., 2020), as shown in Figure 1(*right*).

### 4.3 Ablation studies

**Contribution of each component.** Figure 4a demonstrates the contribution of each component in our method across varying AL rounds: candidate set query (Eq. (5)), cost reduction from $\alpha^*$ (Eq. (7)), and the proposed acquisition function (Eq. (8)). The results show consistent performance improvements from each component in every round. The performance gap between CQ+Ent and CSQ($\alpha = 0.1$)+Ent verifies the efficacy of proposed CSQ framework, which provides the largest improvement. The gap between CSQ($\alpha = 0.1$)+Ent and CSQ+Ent shows the impact of $\alpha$ optimization, offering modest but steady gains across rounds. Finally, the gap between CSQ+Ent and CSQ+Cost(Ent) shows the effectiveness of our acquisition function, particularly from round 4 to 6.

**Impact of calibration set size.** In Figure 4b, we evaluate the relative labeling cost (%) at the fifth round with varying calibration set sizes $n_{\mathrm{cal}}$ in Eq. (4) to assess its impact on the performance on CIFAR-100. As shown in Figure 4b, our method shows consistent performance, varying by less than 2%p as the calibration set size changes from 0.1K to 2K, and significantly outperforms the baseline.

**Detailed ablation study on candidate set design.** Figure 5 illustrates the effectiveness of using conformal prediction (Conformal ($\alpha = 0.1$)) for candidate set construction on CIFAR-100, compared to baselines:

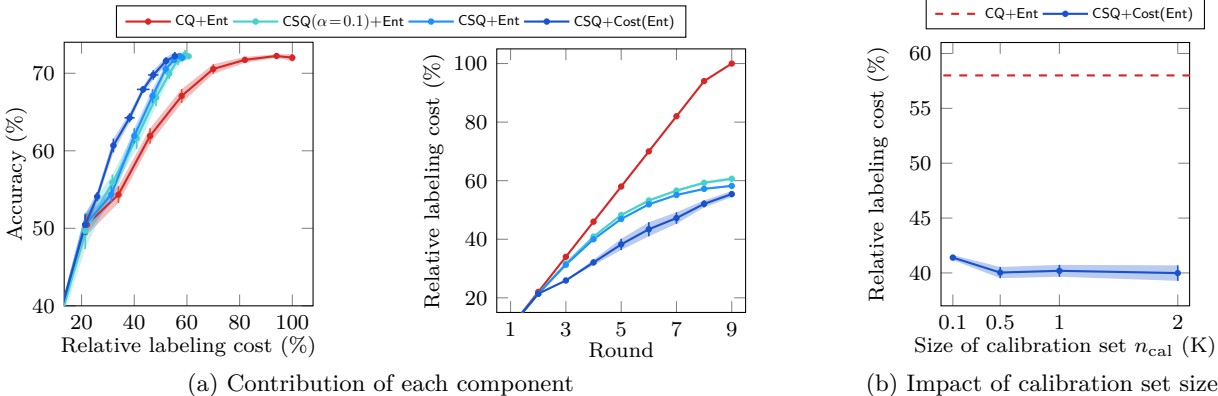

(a) Contribution of each component

(b) Impact of calibration set size

Figure 4: (a) Contribution of each component of our method, measured by accuracy (%) versus relative labeling cost (%) (*left*), and relative labeling cost (%) versus AL rounds (*right*) on CIFAR-100. The results compare the full method (CSQ+Cost(Ent)), the method without acquisition function in Eq. (8) (CSQ+Ent), without $\alpha$ optimization in Eq. (7), where $\alpha$ is fixed to 0.1 (CSQ($\alpha$=0.1)+Ent), and without CSQ (CQ+Ent). All components of our method lead to steady performance improvement over varying rounds. (b) Relative labeling cost (%) at the fifth round with varying calibration set sizes $n_{\mathrm{cal}}$ in Eq. (4) on CIFAR-100. The dashed line indicates the relative labeling cost (%) of CQ+Ent. Our method shows consistent performance with varying calibration set sizes.

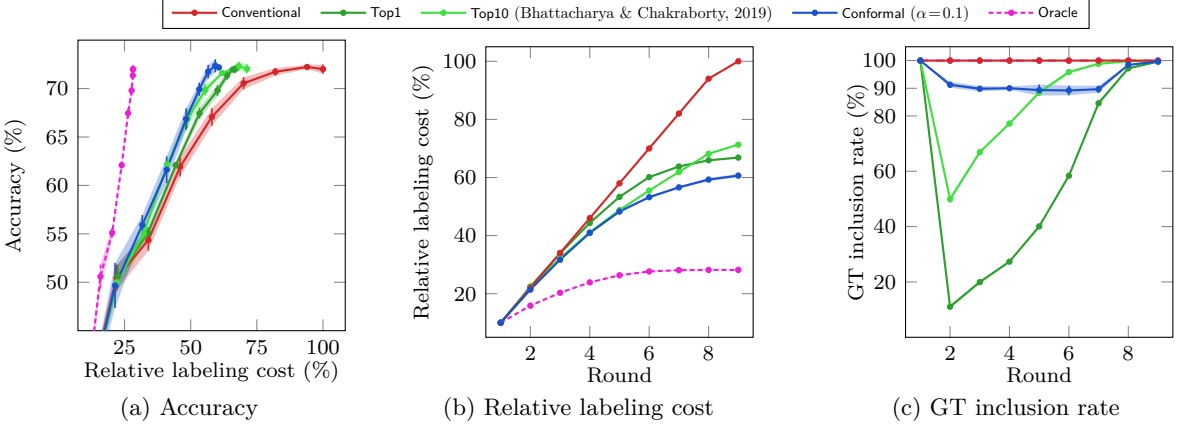

(a) Accuracy

(b) Relative labeling cost

(c) GT inclusion rate

Figure 5: Impact of the candidate set design evaluated on CIFAR-100 using the conventional query with all classes (Conventional), top-1 prediction from model (Top1), top-10 predictions from model (Top10), our method with conformal prediction with fixed $\alpha$ = 0.1 (Conformal($\alpha$=0.1)), and the smallest top-$k$ prediction sets always including the ground-truth class (Oracle). Note that Top10 is a variant of the $n$-ary query (Bhattacharya & Chakraborty, 2019) baseline. For comparison, the same entropy sampling is used to keep the accuracy at each round constant, focusing solely on the labeling cost and isolating the effect of the candidate set design. (a) Our method constantly outperforms the baselines both in accuracy (%) and relative to labeling cost (%). (b) Our design achieves a greater reduction in labeling cost compared to the baselines. (c) Our candidate set effectively includes the ground-truth class in over 90% of cases (= 1 − $\alpha$), even when model accuracy is low.

Conventional (using all classes), Top1 (top-1 prediction), Top10 (top-10 predictions), and Oracle (smallest top-$k$ set always containing the ground truth). Note that Oracle represents an unattainable upper bound requiring knowledge of the ground truth. Top10 is a variant of the $n$-ary query (Bhattacharya & Chakraborty, 2019) baseline. For consistency, we fixed $\alpha$ = 0.1 in Eq. (5). Figures 5a and 5b show that conformal prediction consistently reduces labeling cost compared to the baselines. While Top10 is effective in the early rounds and Top1 becomes more efficient as the model improves, our method adapts and outperforms all baselines in every round. Figure 5c demonstrates that with $\alpha$ = 0.1, our method includes the ground-truth class in over 90% of cases, aligning with Eq. (6), while the top-$k$ baselines show lower inclusion rates, especially in

Table 2: Effectiveness of the proposed cost-efficient sampling (Cost($\cdot$)) on CIFAR-100, reported as *accuracy per cost*. All methods employ the same candidate set query (CSQ) framework, and we simply replace the acquisition function baselines with their Cost($\cdot$) variants. Our sampling consistently improves the performance across various acquisition functions and AL rounds. The best results are highlighted in bold.

| Acquisition | 2nd round | 3rd round | 4th round | 5th round | 6th round | 7th round | 8th round | 9th round |
|---|---|---|---|---|---|---|---|---|
| Ent | 2.36 | 1.74 | 1.55 | 1.43 | 1.36 | 1.30 | 1.26 | 1.24 |
| Cost(Ent) | 2.36 | **2.09** | **1.89** | **1.68** | **1.56** | **1.48** | **1.37** | **1.30** |
| BADGE | 2.51 | 1.94 | 1.69 | 1.51 | 1.40 | 1.34 | 1.29 | 1.27 |
| Cost(BADGE) | **2.64** | **2.17** | **1.92** | **1.75** | **1.58** | **1.49** | **1.37** | **1.32** |
| ProbCover | 2.43 | 1.72 | 1.60 | 1.55 | 1.47 | 1.39 | 1.33 | 1.30 |
| Cost(ProbCover) | 2.43 | **2.10** | **1.98** | **1.79** | **1.66** | **1.52** | **1.39** | **1.32** |
| SAAL | **2.37** | 1.83 | 1.55 | 1.43 | 1.37 | 1.32 | 1.28 | 1.25 |
| Cost(SAAL) | 2.36 | **2.12** | **1.94** | **1.74** | **1.64** | **1.50** | **1.38** | **1.31** |

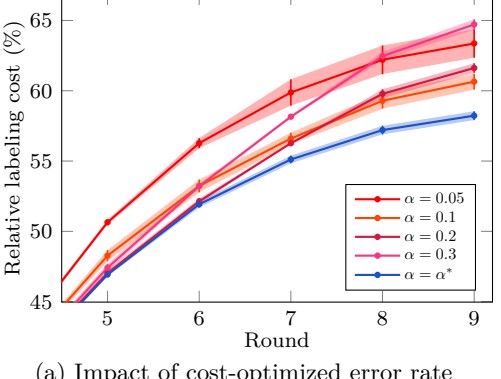
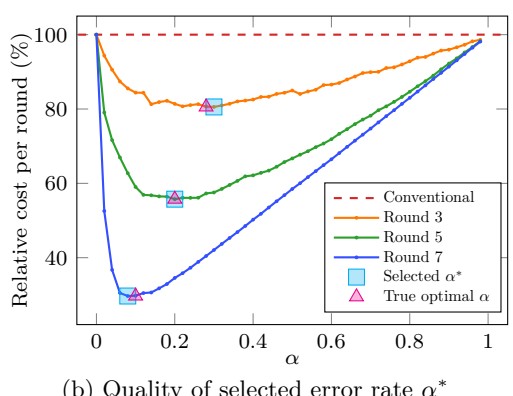

(a) Impact of cost-optimized error rate    (b) Quality of selected error rate $\alpha^*$

Figure 6: Impact of cost-optimized error rate selection as in Eq. (7), evaluated on CIFAR-100 with entropy sampling. (a) Relative labeling cost (%) versus AL rounds with different error rate $\alpha$ compared with the $\alpha^*$ selected by the proposed method (Eq. (7)). This shows our method removes manual tuning by automatically choosing an adaptive error rate at each round. (b) Relative labeling cost per round (%) versus $\alpha$ across various AL rounds. The cost curves reveal that selecting the right error rate greatly lowers labeling cost, with the benefit growing in later rounds. Pink triangles mark the true optimal $\alpha$, blue squares mark $\alpha^*$ from our method. The selected $\alpha^*$ remains consistently close to the true optimum, enabling each round to sharply reduce labeling cost.

early and middle rounds. This demonstrates that conformal prediction effectively adjusts candidate set sizes based on sample uncertainty, ensuring ground-truth inclusion and improving labeling efficiency.

**Detailed ablation study on cost-efficient acquisition function.** In Table 2, we investigate the impact of the proposed cost-efficient sampling (Sec. 3.3) on CIFAR-100, where performance is measured by accuracy per cost (*i.e.*, accuracy divided by relative labeling cost). This metric reflects how efficiently a method achieves high accuracy under a fixed annotation budget—higher values indicate better cost-effectiveness. The proposed cost-efficient sampling strategy consistently improves performance across different acquisition functions, with gains observed throughout various AL rounds. A more detailed accuracy versus labeling cost graph is illustrated in Figure 10.

**Ablation study on cost-optimized error rate selection.** In Figure 6a, we present the impact of cost-optimized error rate selection as in Eq. (7), evaluated on CIFAR-100 using entropy sampling, in terms of relative labeling cost (%). Compared to the baselines using hand-picked error rate values, the cost-optimized error rate $\alpha = \alpha^*$ from the proposed method consistently reduces labeling cost across all AL rounds. This demonstrates that the proposed method reduces the need for manual tuning of the error-rate hyperparameter and instead automatically selects an error rate optimized for each AL round.

### 4.4 In-depth analysis

**Quality of the optimized error rate $\alpha^*$.** In Figure 6b, we compare the optimized error rate ($\alpha^*$, blue squares) selected by our method with the true optimal error rate (pink triangles), showing their relative labeling costs (%) across different AL rounds. As indicated by the labeling cost curves for each error rate, choosing an effective error rate significantly impacts cost reduction, and this impact becomes more prominent at later rounds. Our proposed method adaptively selects an optimized error rate ($\alpha^*$) close to the true optimal error rate at each AL round, substantially reducing labeling costs.

**Analysis on computational complexity.** Figure 7 plots the wall-clock time measured in *minutes* of each strategy as the size of unlabeled pool from ImageNet64x64 grows from 10K to 0.7M images. The dashed curve isolates the cost of the candidate set query (CSQ) itself and reveals a flat line: constructing candidate sets is *constant time*, independent of the unlabeled pool size. All entropy-based samplings, including our proposed CSQ+Cost(Ent), exhibit the expected *linear* slope. This confirms that both the candidate set query and cost-efficient sampling add negligible overhead, preserving the linear complexity of standard entropy sampling. By contrast, CQ+BADGE rises steeply owing to the $k$-means++ seeding inside BADGE[2], even with an accelerated implementation[3].

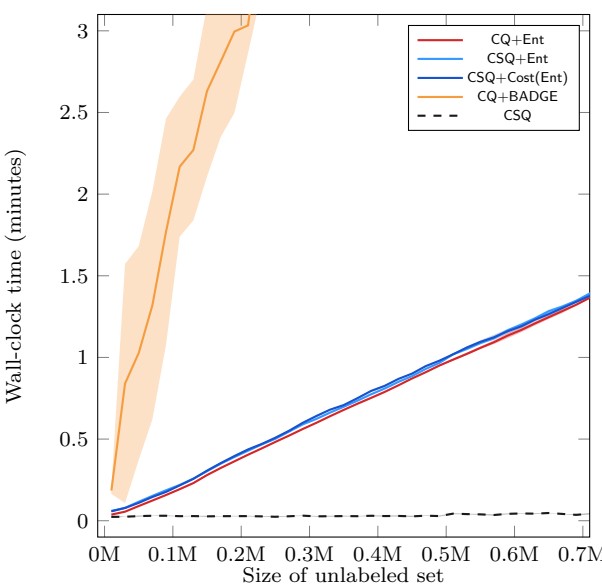

Figure 7: Wall-clock time versus unlabeled pool size on ImageNet64x64. CSQ (dashed) runs in constant time, independent of pool size. Our sampling (Cost(Ent)) adds a negligible overhead, keeping the same linear complexity as plain entropy sampling.

Table 3: Detailed comparison of wall-clock time measured in *minutes* (mean $\pm$ standard deviation across three different random seeds) with an unlabeled data pool of size 1.21 million from ImageNet64x64. The table decomposes the total runtime into *Query* (for constructing candidate sets) and *Sampling* (for selecting informative samples). While conventional querying strategies (CQ+Ent and CQ+BADGE) incur no query-time cost by presenting all classes to the oracle, the proposed CSQ framework introduces a marginal overhead ($<0.05$ minutes), thereby maintaining practical efficiency. Also, our cost-efficient sampling Cost(Ent) maintains the linear time complexity of standard entropy sampling, while adding only marginal additional cost. Together, these results show that the full pipeline CSQ+Cost(Ent) achieves competitive computational efficiency without sacrificing scalability.

|  | Query | Sampling | Total |
|---|---|---|---|
| CQ+Ent | $0.00_{\pm 0.00}$ | $2.29_{\pm 0.01}$ | $2.29_{\pm 0.01}$ |
| CQ+BADGE | $0.00_{\pm 0.00}$ | $17.64_{\pm 0.31}$ | $17.64_{\pm 0.31}$ |
| CSQ+Ent | $0.04_{\pm 0.01}$ | $2.27_{\pm 0.01}$ | $2.31_{\pm 0.01}$ |
| CSQ+Cost(Ent) | $0.04_{\pm 0.00}$ | $2.26_{\pm 0.02}$ | $2.31_{\pm 0.02}$ |

Table 3 gives an absolute breakdown at the largest pool size (1.21 M images). Although conventional queries (CQ+Ent, CQ+BADGE) have zero query cost—they simply show all classes to the oracle—our CSQ adds only $<0.05$ minutes, a negligible overhead that confirms the constant-time trend in the Figure 7. Likewise, replacing plain entropy sampling with its cost-efficient variant (Cost(Ent)) increases sampling time by less than 0.01 min, preserving the same linear complexity. Both components of our pipeline are light—CSQ for querying and Cost($\cdot$) for sampling—so the full method (CSQ+Cost(Ent)) remains scalable even on million-scale pools.

---

[2]For computational feasibility we therefore cap its AL budget at 5000 images and its calibration set size at 500.
[3]We adopt the accelerated code from Zhang et al. (2024).

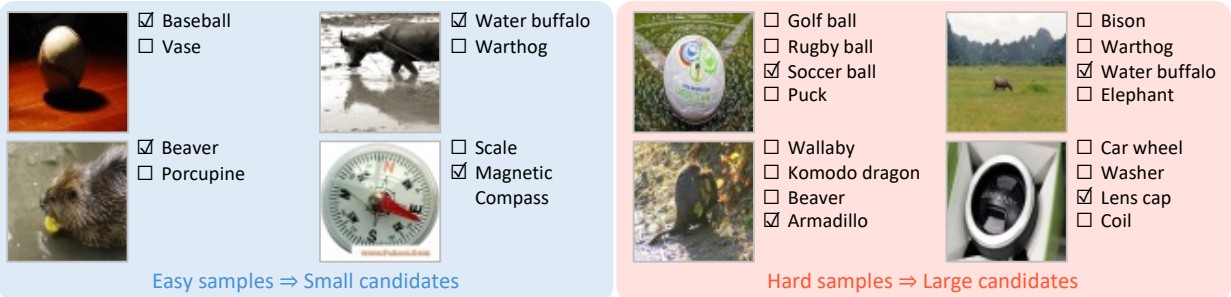

Figure 8: Examples of input images and their corresponding candidate sets constructed using our method at the fifth AL round on ImageNet64x64. For each image, the ✓ indicates the ground-truth class. Our method adjusts the size of each candidate set on the fly: it shrinks the set for *confident* cases (*left*) to reduce annotation cost, and enlarges it for *uncertain* ones (*right*) to include the ground-truth class.

**Examples of constructed candidate sets.** In Figure 8, we present input images and their corresponding candidate sets on ImageNet64x64. Thanks to the conformal prediction, the proposed method allows flexible adjustments of the candidate set for each sample.

## 5    Conclusion

We have introduced candidate set query, an active learning framework that reduces the labeling cost effectively and efficiently by narrowing down the candidate set likely to include the ground-truth class. We have also proposed a novel acquisition function that balances model performance and labeling cost by taking expected candidate set sizes into account. Empirical evaluations on CIFAR-10, CIFAR-100, and ImageNet64x64 confirm the effectiveness of our framework.

**Limitations and future work.** One limitation is that the proposed acquisition function lacks theoretical guarantee for label complexity (Dasgupta, 2011; Hanneke et al., 2014) at this point. Establishing a theoretical understanding to quantify the cost required to achieve a target performance remains an interesting direction for future work.

**Acknowledgement.**   This work was supported by the IITP grants (RS-2021-II212068–25%, RS-2024-00457882–25%, RS-2022-II220926–40%, RS-2019-II191906–10%) funded by Ministry of Science and ICT, Korea.

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

# Appendix

## A    Details of User Study

Q. Select the class that corresponds to the image.

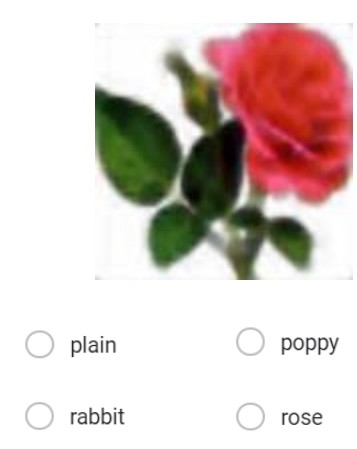

   ○ plain      ○ poppy

   ○ rabbit     ○ rose

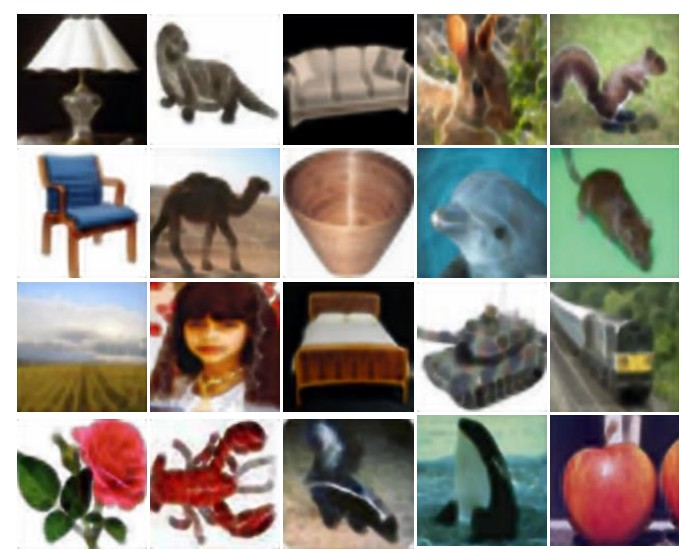

(a) Questionnaire with four candidates ($k = 4$)    (b) Example queries in CIFAR-100

Figure 9: Questionnaire and examples used in the user study. (a) Each question contains an instruction, an image, and a set of candidates. In this case, the candidate set size is 4. (b) We utilize 20 images from CIFAR-100, each with a resolution of 128 x 128 pixels.

We conduct a user study to examine how the size of a candidate set, $k$ in Sec. 3.1, affects the annotation time in practice. Figure 9 presents examples of the questionnaire and all images used in our user study. To facilitate easy comparison with the theoretical costs (Hu et al., 2018), we set the candidate set sizes to 4, 8, 16, and 32. To be specific about Figure 9, we use CIFAR-100 images resized to $128 \times 128$ using super resolution[4] to enhance visibility for annotators. We first randomly select 20 classes in CIFAR-100 and choose one image per class to organize the questionnaires. For small-sized candidate sets, we ensure the inclusion of the ground truth by randomly trimming around it when generating the candidate sets.

We divide 44 annotators into four groups of 11 for each candidate set size to perform labeling tasks. To account for potential outliers, we exclude the results of the annotators whose time taken deviates the most from the average time in each group. Table 4 shows that as the candidate set size increases, the time per query increases and the accuracy decreases. In addition, on the right side of Table 4, the comparison between the experimental costs and theoretical costs reveals a significant correlation of 0.97.

Table 4: User study for different sizes of candidate set query.

| $k$ | Total time (s) | Time per query (s) | Accuracy (%) | Experimental | Theoretical |
|---|---|---|---|---|---|
| 4 | $\mathbf{69.4}_{\pm 13.8}$ | $\mathbf{3.47}_{\pm 0.69}$ | $\mathbf{100.0}_{\pm 0.0}$ | 2.0 | 2 |
| 8 | $91.5_{\pm 27.3}$ | $5.20_{\pm 1.36}$ | $98.5_{\pm 3.2}$ | 2.6 | 3 |
| 16 | $116.9_{\pm 29.6}$ | $6.94_{\pm 1.48}$ | $99.5_{\pm 1.5}$ | 3.4 | 4 |
| 32 | $166.9_{\pm 30.8}$ | $8.35_{\pm 1.54}$ | $95.5_{\pm 5.2}$ | 4.8 | 5 |

---

[4]https://www.kaggle.com/datasets/joaopauloschuler/cifar100-128x128-resized-via-cai-super-resolution

## B  Implementation Details and Configuration

Table 5 presents the configuration of our main experiments for each dataset. In all experiments, we fixed the per-round budget, which limits the number of annotated instances per active learning (AL) round. Given this budget constraint, we compute the labeling cost for each AL round to assess labeling efficiency. The batch size for CIFAR-10, CIFAR-100, and ImageNet64x64 was determined to be 128. We normalized the input image to ensure the stability of the training. We trained our classification model on CIFAR-10 and CIFAR-100 using NVIDIA RTX 3090 and on ImageNet64x64 using 4 NVIDIA A100 GPUs in parallel. The training requires about 5 GPU hours for CIFAR-10 and CIFAR-100, and about 1.5 GPU days for ImageNet64x64.

Table 5: Detailed dataset and budget configuration for the proposed scenario.

| Dataset | $L$ | $\log_2 L$ | Size | Cost of full label | # of rounds | Per-round budget |
|---|---|---|---|---|---|---|
| CIFAR-10 | 10 | 3.322 | 50K | 166.1K | 10 | 6K |
| CIFAR-100 | 100 | 6.644 | 50K | 332.2K | 9 | 6K |
| ImageNet64x64 | 1000 | 9.966 | 1.2M | 12.7M | 16 | 60K |

## C  Additional Clarification on Candidate Set Construction

**The detailed procedure of computing $\hat{Q}(\alpha)$ in Eq. (4).** We begin with computing the collection of conformal scores $\mathbf{s}$ for the calibration dataset $\mathcal{D}_{\mathrm{cal}}$. For each data point $(\mathbf{x}_i, y_i) \in \mathcal{D}_{\mathrm{cal}}$, the conformal score is defined as:

$$s_i := 1 - P_{\boldsymbol{\theta}}(y_i \mid \mathbf{x}_i), \quad \text{for } i = 1, 2, \cdots, n_{\mathrm{cal}} , \tag{9}$$

where $n_{\mathrm{cal}} = |\mathcal{D}_{\mathrm{cal}}|$. Using these scores, we define the empirical distribution function $F_n(s)$, which measures the proportion of scores less than or equal to a given value $s$. Formally, $F_n(s)$ is expressed as:

$$F_n(s) = \frac{1}{n_{\mathrm{cal}}} \sum_{i=1}^{n_{\mathrm{cal}}} \mathbb{1}[s_i \leq s] , \tag{10}$$

where $\mathbb{1}[\cdot]$ is an indicator function. The $(1 - \alpha)$ empirical quantile is then defined as the smallest score $s_i$ such that the proportion of scores satisfying $s_i \leq s$ is at least $(1 - \alpha)$. Mathematically, this is given as $\min_{i \in [n_{\mathrm{cal}}]} \{F_n(s_i) \geq 1 - \alpha\}$, where $[n_{\mathrm{cal}}] = \{1, 2, \cdots, n_{\mathrm{cal}}\}$.

$$\hat{Q}(\alpha) := \min_{i \in [n_{\mathrm{cal}}]} \{F_n(s_i) \geq (1 - \alpha)\} . \tag{11}$$

Note that Eq. (11) is equivalent to Eq. (4).

## D  Impact of Cost-efficient Sampling across Sampling Strategies

In Figure 10, we compare different sampling strategies when combined with their cost-efficient variants on CIFAR-100. Notably, the performance of cost-efficient sampling consistently improves when combined with a range of acquisition functions. These results show that our cost-efficient acquisition method (Eq. (8)) performs consistently well when paired with four diverse acquisition strategies: entropy sampling, BADGE (Ash et al., 2020), ProbCover (Yehuda et al., 2022), and SAAL (Kim et al., 2023). This suggests that it can be readily combined with a broad range of acquisition functions beyond those evaluated in our experiments.

## E  Discussion on Handling Outliers and Anomalous Data Points

Dealing with out-of-distribution (OOD) data points showing high uncertainty scores has been a chronic issue in active learning and may affect the efficiency of candidate set query (CSQ). Recent open-set active

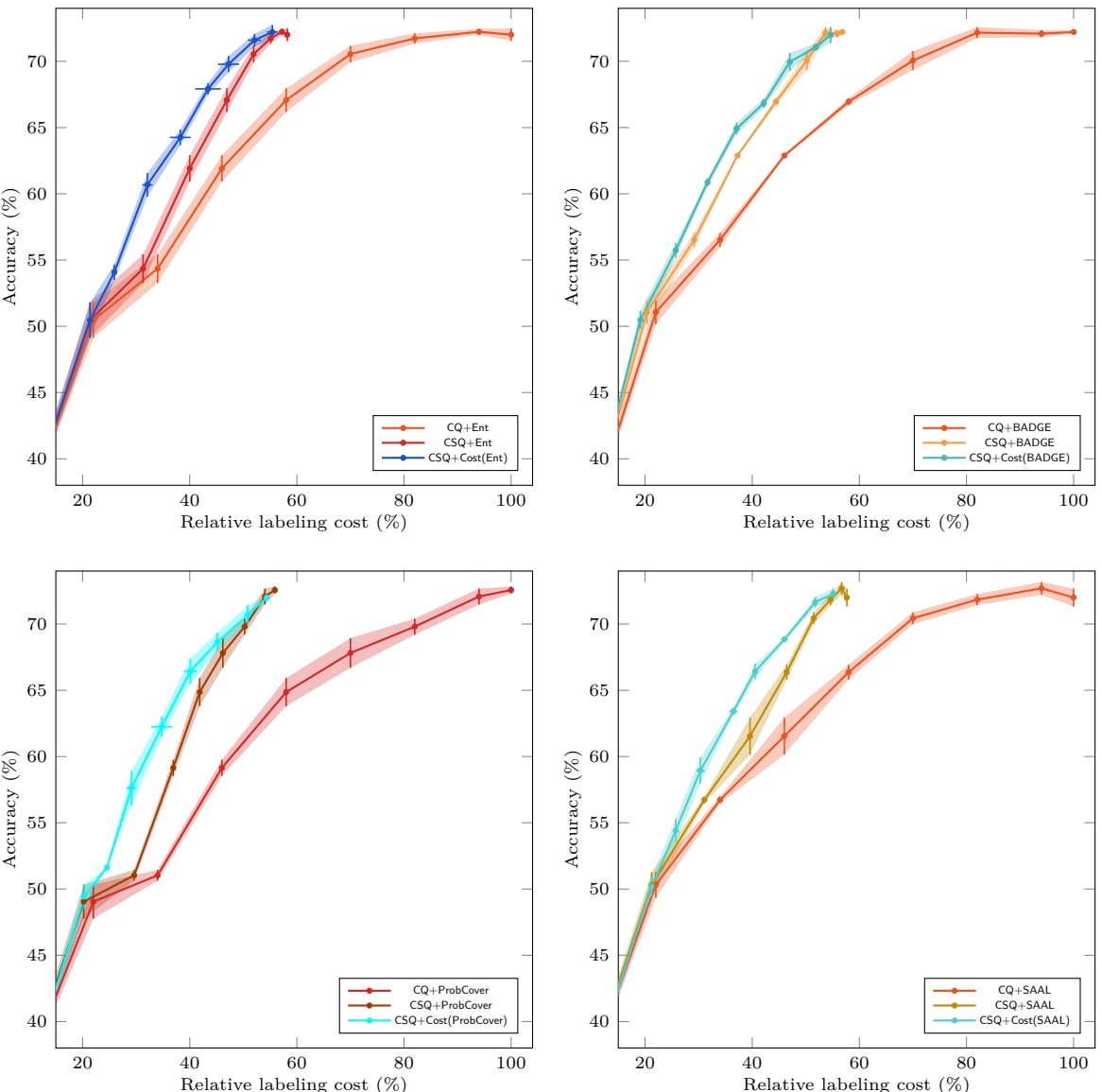

Figure 10: Comparison of different sampling methods and their cost-sampling variants on CIFAR-100. Each subplot shows a triple of corresponding methods.

learning approaches (Du et al., 2021; Kothawade et al., 2021; Ning et al., 2022; Park et al., 2022; Yang et al., 2024) tackle this by filtering out OOD samples during active sampling using an OOD classifier. Our CSQ framework integrates seamlessly with these methods, focusing on labeling in-distribution (ID) samples to prevent cost inefficiency.

However, as OOD classifiers are not flawless, some OOD samples may still be selected. One advantage of our method is its ability to leverage the calibration set to capture information about such mixed OOD samples. This enables adjustments such as increasing the OOD classifier threshold to exclude more OOD-like data or incorporating the OOD ratio into the alpha optimization process in Eq. (7). Optimizing the combination of OOD and ID classifier scores within the calibration set or designing better OOD-aware queries presents promising future research directions.

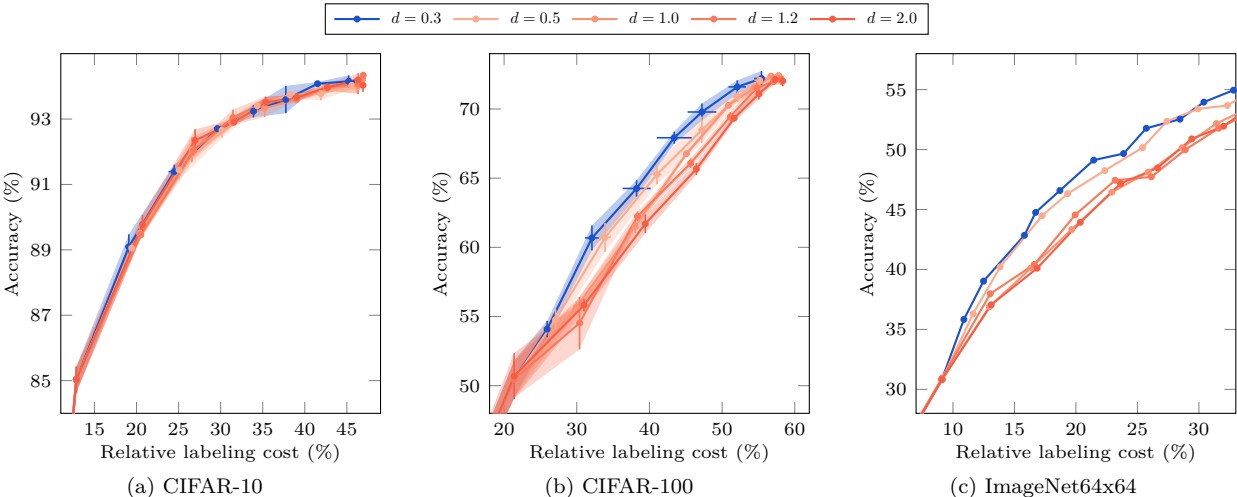

Figure 11: Accuracy (%) versus relative labeling cost (%) with varying hyperparameter $d$ in Eq. (8) across AL rounds, evaluated on CIFAR-10, CIFAR-100 and ImageNet64x64 with CSQ+Cost(Ent). We set $d = 0.3$ for all datasets in our main experiments.

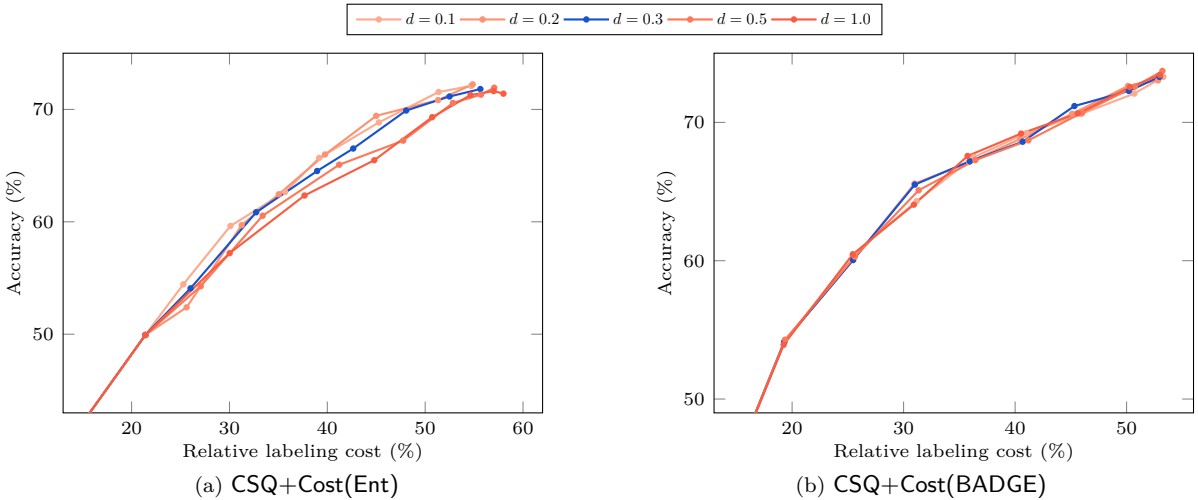

Figure 12: Accuracy (%) versus relative labeling cost (%) with varying hyperparameter $d$ in Eq. (8) across AL rounds on CIFAR-100. (a) cost-efficient sampling combined with entropy sampling (CSQ+Cost(Ent)). (b) cost-efficient sampling combined with BADGE (CSQ+Cost(BADGE)).

## F   Impact of Informativeness-Cost Balancing Hyperparameter $d$

The hyperparameter $d$ in our acquisition function (Eq. (8)) balances the trade-off between labeling cost and the informativeness of a sample. We provide a comprehensive analysis showing the trend of performance in accuracy with varying $d$ values over AL rounds across datasets (Figure 11) and sampling strategies (Figure 12), and additionally demonstrate the results in the language domain (Figure 13).

For CIFAR-10 (Figure 11a), both accuracy and labeling cost remain robust to the change of $d$, varying only 0.5%p in accuracy. For CIFAR-100 and ImageNet64x64 (Figure 11b and Figure 11c) combined with entropy-based sampling, the overall performance improves as $d$ decreases. On the other hand, as shown in Figure 12 and Figure 13, the accuracy and labeling cost exhibit no clear correlation with $d$. Nevertheless, the performance and cost-efficiency remain insensitive across different datasets, domains, and sampling strategies. This suggests that $d$ can be easily selected without extensive tuning. We fix $d = 0.3$ in all main experiments to demonstrate the robustness of our method without dataset-specific adjustment.

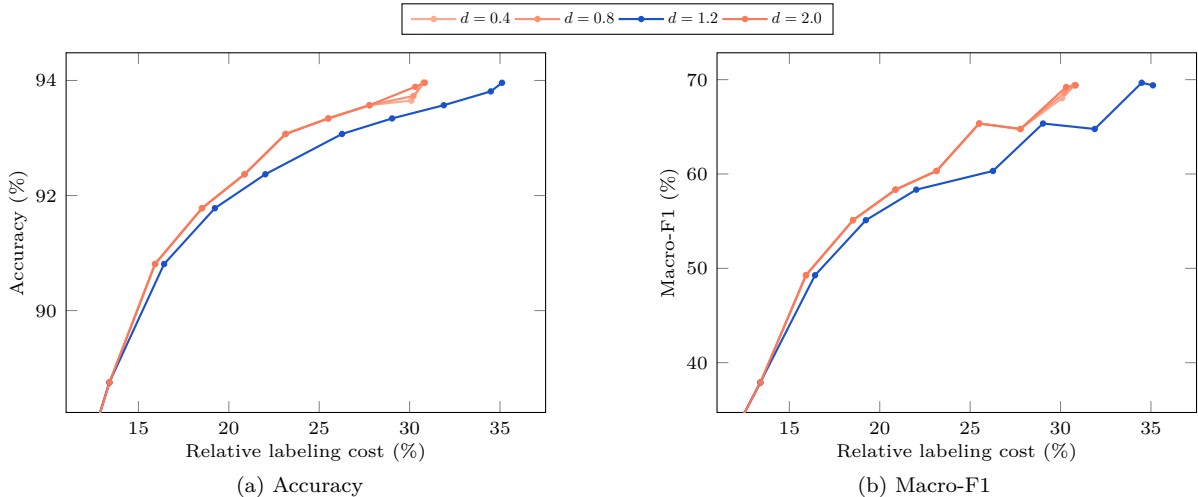

(a) Accuracy

(b) Macro-F1

Figure 13: (a) Accuracy (%) and (b) Macro-F1 (%) versus relative labeling cost (%) with varying hyperparameter $d$ in Eq. (8) across AL rounds on R52 with CSQ+Cost(Ent).

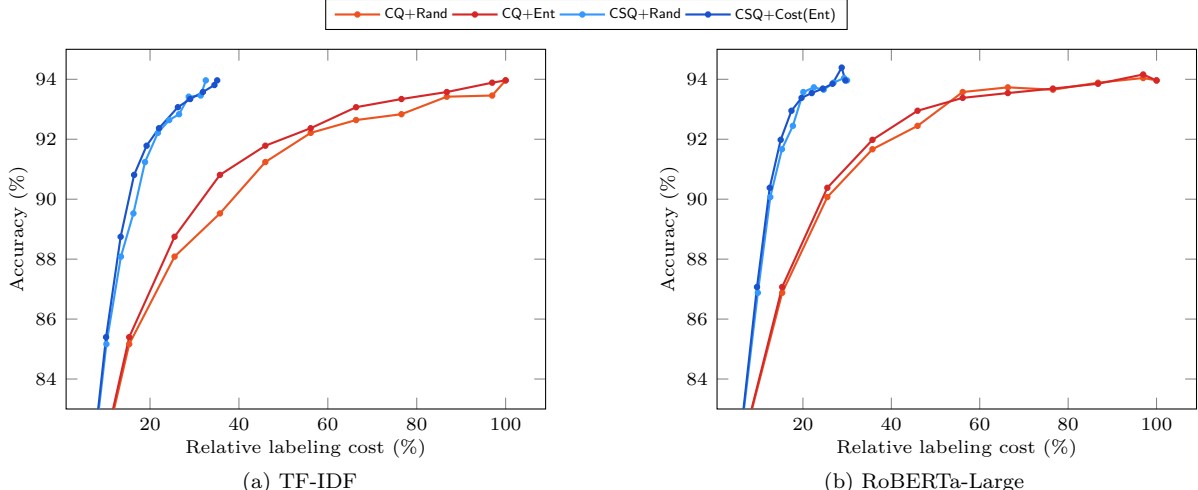

(a) TF-IDF

(b) RoBERTa-Large

Figure 14: Comparison between conventional query (CQ) and candidate set query (CSQ) with random sampling (Rand), entropy sampling (Ent), and cost-efficient entropy sampling (Cost(Ent)) on text classification task with R52 dataset. We adopt (a) TF-IDF (Manning, 2009) and (b) RoBERTa-Large (Liu et al., 2019b) as feature extractors. We report the results as Accuracy (%) versus relative labeling cost (%). CSQ approaches (blue lines) consistently outperform the CQ baselines (red lines) by a significant margin across various budgets and acquisition functions.

## G    Experiments in Language Domain

**Dataset.** The R52 dataset (Lewis, 1997) is a subset of the Reuters-21578 (Lewis, 1997) news collection, specifically curated for text classification tasks. It comprises documents categorized into 52 distinct classes, with a total of 9,130 documents. The dataset is divided into 6,560 training documents and 2,570 testing documents. Each document is labeled with a single category, and the categories are selected to ensure that each has at least one document in both the training and testing sets. This structure makes the R52 dataset particularly suitable for evaluating text classification models.

**Implementation details.** We adopt an SVM classifier (Cortes, 1995) with a sigmoid kernel, evaluating with two feature extractors: TF-IDF (Manning, 2009) and RoBERTa-Large (Liu et al., 2019b). We conduct 11 AL rounds of consecutive data sampling and model updates, where the per-round budget is 600. The

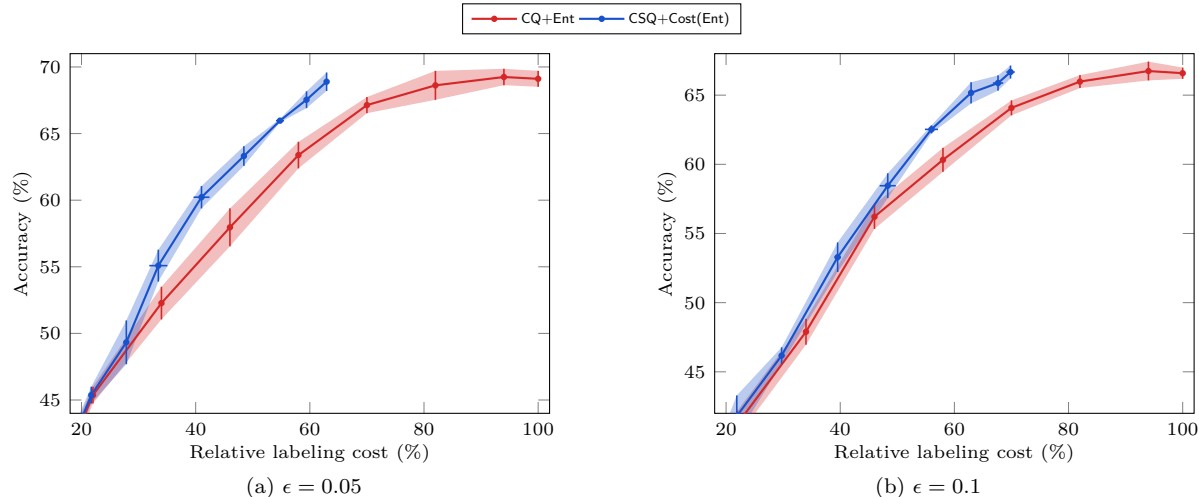

Figure 15: Comparison between conventional query (CQ) and candidate set query (CSQ) with entropy sampling (Ent) and the proposed acquisition function with entropy measure (Cost(Ent)) on CIFAR-100 with label noise across AL rounds with varying noise level: (a) Noise rate of 0.05. (b) Noise rate of 0.1. The proposed CSQ+Cost(Ent) consistently outperforms CSQ+Ent across various AL rounds and noise rates.

hyperparameter $d$ for our acquisition function is set to 1.2. In the initial round, we randomly sample 300 samples. In each round, the model is evaluated based on two factors: accuracy (%) and relative labeling cost (%).

Figure 14 presents a comparison of candidate set query (CSQ) and conventional query (CQ) on the text classification dataset (R52) with random sampling (Rand), entropy sampling (Ent), and our acquisition function with entropy measure (Cost(Ent), Eq. (8)) across AL rounds. CSQ approaches consistently outperform the CQ baselines by a significant margin across various budgets and acquisition functions. Especially at round 10 with the TF-IDF feature extractor, CSQ+Rand reduces labeling cost by 65.6%p compared to its conventional query baseline. The result demonstrates that the proposed CSQ framework generalizes to the text classification domain.

## H  Experiments on Real-World Datasets

**Experiment on datasets containing label noise.** We evaluate the candidate set query (CSQ) framework on CIFAR-100 with noisy labels, simulating a scenario where human annotators misclassify images into random classes with a noise rate $\epsilon$. This is modeled using a uniform label noise (Frénay & Verleysen, 2013) with $\epsilon$ set to 0.05 and 0.1. Note that this scenario is unfavorable for CSQ, as a misclassifying annotator would reject the actual true label even if the candidate set includes it. Figure 15 compares CSQ and conventional query (CQ) on CIFAR-100 with noisy labels using entropy sampling (Ent) and our acquisition function with entropy measure (Cost(Ent)) across 2, 6, and 9 rounds.

Despite the disadvantageous scenario, our method (CSQ+Cost(Ent)) reduces labeling cost compared to the baseline (CQ+Ent) across varying AL rounds and noise rates. At round 9, CSQ+Cost(Ent) achieves cost reductions of 33.4%p and 27.4%p at noise rates of 0.05 and 0.1, respectively. It also consistently outperforms the baseline in terms of accuracy per labeling cost, demonstrating the robustness of CSQ. Additionally, CSQ has the potential to reduce label noise, as narrowing the candidate set can lead to more precise annotations. Our user study (Table 1) shows that reducing candidate set size improves annotation accuracy, suggesting that CSQ can further enhance performance by reducing label noise.

**Experiment on datasets containing class imbalances.** Figure 16 compares candidate set query (CSQ) and conventional query (CQ) on CIFAR-100-LT (Cui et al., 2019), a class-imbalanced version of CIFAR-100, using entropy sampling (Ent), and our acquisition function with entropy measure (Cost(Ent)) across AL

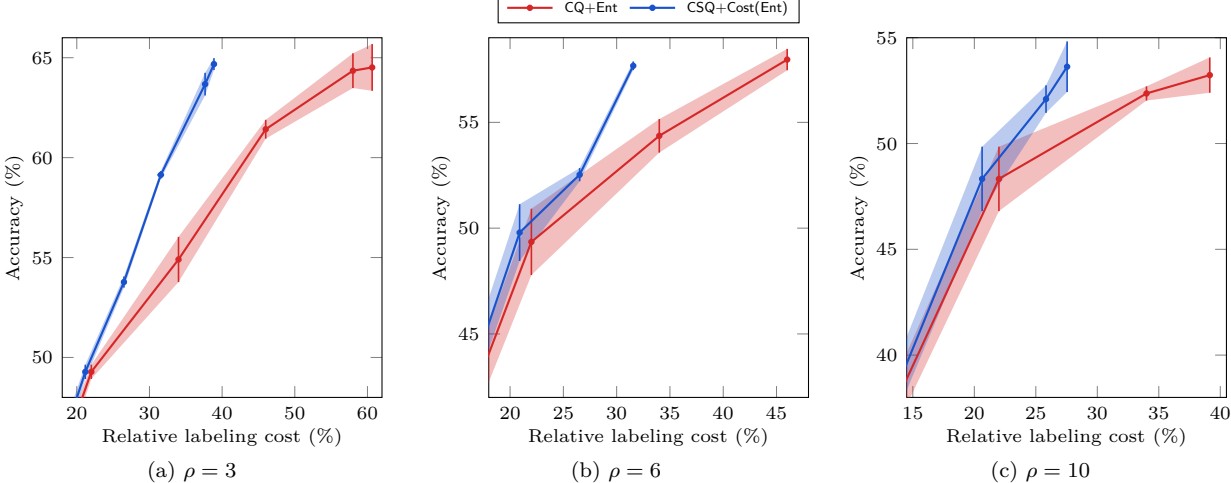

(a) $\rho = 3$         (b) $\rho = 6$         (c) $\rho = 10$

Figure 16: Comparison between conventional query (CQ) and candidate set query (CSQ) with entropy sampling (Ent) and the proposed acquisition function with entropy measure (Cost(Ent)) on CIFAR-100-LT, a variant of CIFAR-100 with class imbalance, across AL rounds with varying imbalance level: (a) Imbalance ratio of 3. (b) Imbalance ratio of 6. (c) Imbalance ratio of 10. The proposed approach (CSQ+Cost(Ent)) consistently outperforms the baseline (CSQ+Ent) across various AL rounds and noise rates. Note that the maximum AL rounds vary with the imbalance ratio due to dataset size, with a maximum of 4 rounds for ratios of 3 and 6, and 6 rounds for a ratio of 10.

rounds. The experiments use imbalance ratios (*i.e.*, ratios between the largest and smallest class sizes) of 3, 6, and 10. Note that the maximum AL rounds vary with the imbalance ratio due to dataset size, with a maximum of 4 rounds for ratios of 3 and 6, and 6 rounds for a ratio of 10.

The result shows that our method (CSQ+Cost(Ent)) reduces labeling cost compared to the baselines (CQ+Ent) by significant margins across varying AL rounds and imbalance ratios. Specifically, at round 4, CSQ+Cost(Ent) achieves cost reductions of 31.1%p and 29.2%p at imbalance ratios of 6 and 10, respectively. In terms of accuracy per labeling cost, CSQ+Cost(Ent) consistently outperforms the baseline, demonstrating the robustness of the CSQ framework in class-imbalanced scenarios.

## I    Experiments on Products-10K

To evaluate the scalability of CSQ with respect to the number of classes, we conduct experiments on the Products-10K (Bai et al., 2020) dataset, which contains 10K classes. We perform 6 AL rounds of consecutive data sampling and model updates. For the first round, 30K images are used to construct the initial training dataset, and the per-round budget is set to 22K. The calibration dataset is built by randomly selecting 5K images at each round.

We compare CSQ combined with cost-efficient entropy sampling (CSQ+Cost(Ent)) and the conventional query baseline (CQ+Ent) evaluated using three model architectures: EfficientNet-B3 (Tan & Le, 2019), ResNet50 (He et al., 2016), and ViT-S/16 (Dosovitskiy et al., 2021). ResNet50 and ViT-S/16 are DINO-pretrained on ImageNet-1K (Deng et al., 2009) to further assess the compatibility of CSQ with advanced pre-trained models, while EfficientNet-B3 is trained from scratch.

As shown in Figure 17, CSQ consistently achieves higher performance per unit cost compared to the CQ baseline across model architectures and training strategies. In particular, for EfficientNet-B3, CSQ reduces labeling cost by 27%p without any performance degradation, demonstrating that the proposed method scales effectively to scenarios with a huge number of classes.

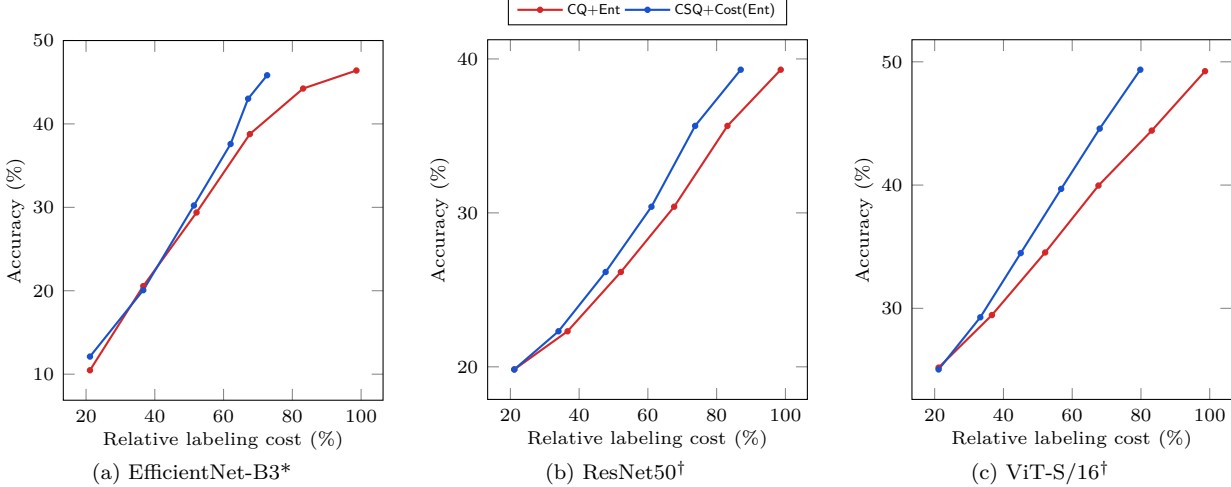

Figure 17: Comparison between conventional query combined with entropy sampling (CQ+Ent) and candidate set query combined with cost-efficient entropy sampling (CSQ+Cost(Ent)) on Products-10K (Bai et al., 2020). We use (a) EfficientNet-B3 (Tan & Le, 2019), (b) ResNet50 (He et al., 2016) model, and (c) ViT-S/16 (Dosovitskiy et al., 2021) as backbones. * and † indicate the models are trained from scratch and DINO-pretrained on ImageNet-1K (Deng et al., 2009), respectively. Fine-tuned models are evaluated on Products-10K via linear probing.

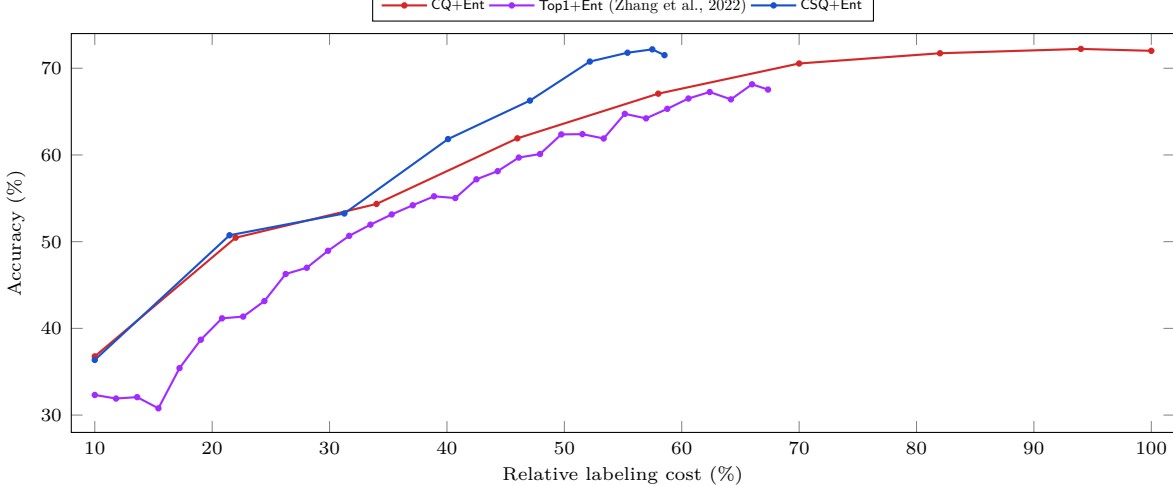

Figure 18: Comparison of conventional query (CQ), top-1 query (Top1) (Zhang et al., 2022), and candidate set query (CSQ) combined with entropy sampling on CIFAR-100. Top1 asks the oracle whether the top-1 prediction is the ground-truth class, potentially yielding partial labels. For a fair comparison, we employ negative learning loss (Kim et al., 2019) to utilize samples with partial labels. Also, samples with partial labels are sampled with replacement until they get full labels.

## J    Comparison with Weakly Supervised Baseline

As described in Sec. 3.1, the oracle is prompted with a follow-up query only when the ground-truth label is absent from the candidate set. To assess the effectiveness of CSQ's two-query design, we compare three types of query strategies: the conventional query (CQ), a weakly supervised baseline (Zhang et al., 2022) that constructs candidate sets solely from top-1 predictions (*i.e.*, pseudo labels) (Top1), and our candidate set query (CSQ). Unlike CSQ, Top1 does not issue a second query, leaving incorrectly pseudo-labeled samples as partial labels, requiring specialized negative learning loss for training.

As shown in Figure 18, `CSQ+Ent` achieves the highest performance among all methods. Despite leveraging partial label learning (Kim et al., 2019), `Top1+Ent` fails to match CSQ's accuracy, as negative learning loss introduces noisy signals in early rounds. Since CSQ produces no partial labels, `CSQ+Ent` matches the performance of `CQ+Ent` while improving cost-efficiency. Moreover, since CSQ does not require a specialized loss function, it can be seamlessly integrated into a conventional classification framework.

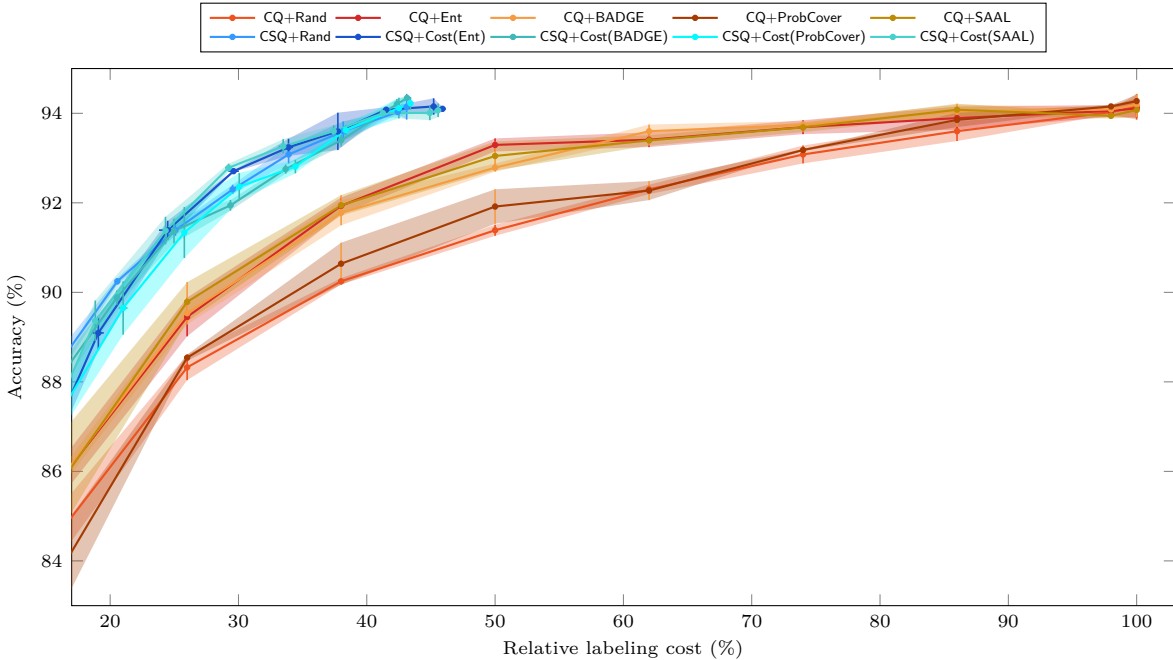

Figure 19: Accuracy (%) versus relative labeling cost (%) on CIFAR-10.

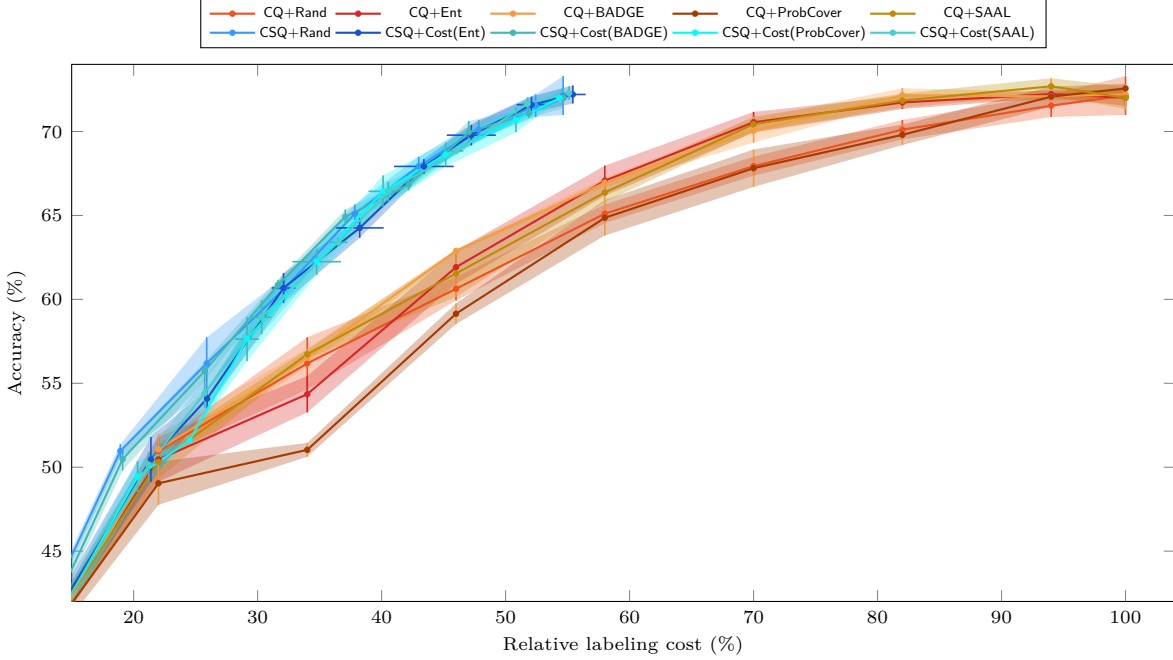

Figure 20: Accuracy (%) versus relative labeling cost (%) on CIFAR-100.

