# OpenReview forum: "Enhancing Cost Efficiency in Active Learning with Candidate Set Query"
_TMLR — Accepted by TMLR_

### Review · Reviewer_opMR · 2025-05-28

**Summary Of Contributions:**

This work presents CSQ, which seeks to minimize labeling costs by presenting annotators with a subset of likely classes rather than the full set. By leveraging conformal prediction, the model dynamically generates small, reliable candidate sets that adapt over successive AL rounds. An acquisition function is introduced to prioritize data points that offer high information gain at lower costs. Empirical evaluations on datasets such as CIFAR-10, CIFAR-100, and ImageNet64x64 demonstrate the effectiveness and scalability of the framework.

**Audience:**

Yes

**Claims And Evidence:**

Yes

**Requested Changes:**

See weaknesses.

**Strengths And Weaknesses:**

# Strengths
1.	The CSQ framework introduces a practical method to reduce annotator workload by narrowing down the class options.
2.	Utilizing conformal prediction allows the system to adaptively generate candidate sets that balance reliability and size, enhancing the efficiency of the annotation process.

# Weaknesses
1.	The paper does not extensively discuss how annotation errors within the candidate sets might impact the model's performance or how the system handles such errors. For example, in the case when the Oracle model still has a chance to give the wrong candidates, how would the proposed method perform?
2.	The framework assumes the availability of an oracle to validate the candidate sets, which may not always be feasible in real-world scenarios, especially where expert annotators are scarce.
3.	While the method is tested on datasets like ImageNet64x64, the scalability, and effectiveness of CSQ in scenarios with an even larger number of classes remain unaddressed.
4.	The paper could benefit from a more comprehensive comparison with other active learning strategies, particularly those that also aim to reduce labeling costs or utilize weak supervision.
5.	The absence of publicly available code or detailed implementation details may hinder reproducibility and adoption by the broader research community.

---

> ### Author Response · Authors · 2025-06-25
> **Response to Reviewer opMR (Part 1/2)**
>
> ### **General reply**
> We sincerely appreciate your thorough review and valuable feedback. We provide additional explanations and clarifications to address your concerns below. If there are any remaining concerns, please let us know, and we will gladly provide further clarification.
>
> ---
>
> ### **Weakness 1. Label noise during annotation**
> > The paper does not extensively discuss how annotation errors within the candidate sets might impact the model's performance or how the system handles such errors. For example, in the case when the Oracle model still has a chance to give the wrong candidates, how would the proposed method perform?
>
> Thank you for your feedback.
>
> To demonstrate the robustness of candidate set query (CSQ) under label noise, we evaluate two types of annotation error scenarios: (1) label noise over the complete set of classes and (2) label noise within the candidate set.
>
> **Label noise over the complete set of classes.** As discussed in Section I, we simulate symmetric label noise by randomly flipping labels to other classes in the complete set (5% and 10% rates). As shown in Figure 14, CSQ maintains the same accuracy as the conventional query (CQ) baseline while reducing labeling costs by 35%p, indicating its robustness to global noise.
>
> **Label noise within the candidate set (i.e., the example case in your comment).** We simulate noise by randomly selecting an incorrect label from the candidate set with 5% or 10% probability. Figure 18 in the revised manuscript shows that CSQ still outperforms CQ under these conditions, reducing labeling cost by up to 40%p.
>
> Since the candidate sets constructed using CSQ are often small and likely to include the ground-truth label, the oracle rarely selects the wrong label within the candidate set. Even if an incorrect label is chosen, that error is limited to semantically related classes drawn from the model’s top-$k$ predictions, making such noise less harmful than mistakes drawn from the full set of classes. Additionally, our user study (Table 4) shows that smaller candidate sets not only reduce annotation time but also increase labeling accuracy, suggesting CSQ’s potential to mitigate oracle mislabeling.
>
> ---
>
> ### **Weakness 2. Availability of an oracle**
> > The framework assumes the availability of an oracle to validate the candidate sets, which may not always be feasible in real-world scenarios, especially where expert annotators are scarce.
>
> Our candidate set query (CSQ) framework adopts the same assumption as conventional active learning (AL): the availability of an oracle (i.e., annotator) who labels selected samples.
> This is a standard premise in supervised learning, and AL seeks to reduce annotation cost by asking the oracle to label more informative samples first.
>
> Aside from this, CSQ introduces no further assumptions about oracle availability or capabilities. As described in Section 3.1, CSQ presents the oracle with a narrowed set of candidate labels based on model predictions and asks for the correct label. Only if the ground-truth label is absent from this set is the oracle asked to select from the remaining labels. This two-step process simply reduces annotation effort utilizing *the same* oracle capabilities as in the traditional AL. In other words, **we do not introduce a separate oracle to validate the candidate set.**
>
> We hope this clarification addresses your concern. Please don’t hesitate to let us know if further elaboration would be helpful. We appreciate your insightful feedback.
>
> ---
>
> ### **Weakness 3. Scalability of CSQ**
> > While the method is tested on datasets like ImageNet64x64, the scalability, and effectiveness of CSQ in scenarios with an even larger number of classes remain unaddressed.
>
> Thank you for your valuable feedback.
>
> To evaluate the scalability of candidate set query (CSQ) with respect to the number of classes, we conduct experiments on the Product-10K dataset, which contains 10,000 classes. Figure 19 in the revised manuscript shows the results of CSQ and the baseline evaluated using three model architectures: EfficientNet-B3, ResNet50, and ViT-S/16. EfficientNet-B3 is trained on Products-10K from scratch, while the others are pretrained on ImageNet-1K in a self-supervised manner and evaluated on Products-10K via linear probing.
>
> As shown in the figure, CSQ consistently achieves higher performance per unit cost compared to the CQ baseline. In particular, for EfficientNet-B3, CSQ reduces labeling cost by 27%p without any performance degradation, demonstrating that the proposed method scales effectively to scenarios with a huge number of classes.
>
> **References**
> - Yalong Bai, Yuxiang Chen, Wei Yu, Linfang Wang, Wei Zhang. Products-10K: A Large-scale Product Recognition Dataset. 2020.

---

> ### Author Response · Authors · 2025-06-25
> **Response to Reviewer opMR (Part 2/2)**
>
> ### **Weakness 4. Comparison with other active learning strategies using weak supervision**
> > The paper could benefit from a more comprehensive comparison with other active learning strategies, particularly those that also aim to reduce labeling costs or utilize weak supervision.
>
> Thank you for raising this valuable point.
>
> In Figure 5, we comprehensively compare our CSQ with the $n$-ary query baselines, which aim to reduce the labeling costs using top-$n$ predictions (Bhattacharya & Chakraborty, 2019), denoted as ‘Top1’ and ‘Top10’. CSQ demonstrates clear superiority due to its novel use of conformal prediction. We will include the n-ary query citation (Bhattacharya & Chakraborty, 2019) in the ‘Top10’ legend of Figure 5.
>
> In addition, Figure 21 in the revised manuscript compares CSQ with a recent active learning method based on weak supervision (Zhang et al., 2022Ω) on CIFAR-100. As shown in the figure, CSQ achieves higher performance while reducing labeling cost by up to 20%p without performance degradation. This result highlights the effectiveness of our conformal prediction-based query design compared to existing query strategies.
>
>
> **References**
> - Aditya R Bhattacharya and Shayok Chakraborty. Active learning with n-ary queries for image recognition. WACV, 2019.
>
> - Yuhang Zhang, Xiaopeng Zhang, Lingxi Xie, Jie Li, Robert C. Qiu, Hengtong Hu, Qi Tian. One-bit Active Query with Contrastive Pairs. CVPR, 2022.
>
> ---
>
> ### **Weakness 5. Availability of code**
> > The absence of publicly available code or detailed implementation details may hinder reproducibility and adoption by the broader research community.
>
> To support reproducibility, we have included the source code and dataset preprocessing scripts in the supplementary material. We hope these resources will facilitate further research and broader adoption.

---

### Review · Reviewer_jH6f · 2025-05-30

**Summary Of Contributions:**

The authors proposed a cost-efficient active learning framework. The algorithm begins by estimating the labeling cost for each unlabeled sample, and then computes an acquisition score. This score is determined by the ratio of the estimated information gain to the estimated labeling cost. The information gain is quantified using established methods, such as entropy, BADGE, or SAAL. The expected cost calculation depends on whether the correct label is included in the candidate set or not. Specifically, the annotator is presented with a label candidate set, where the ground-truth class is included with probability 1-\alpha^*. This candidate set is constructed based on conformal prediction [Romano et al., 2020]. The experiments in Figure 1 shows that with proposed framework, existing methods achieve high accuracy with much less cost.

**Audience:**

Yes

**Claims And Evidence:**

No

**Requested Changes:**

The discussion between the cost and the actual number of labeled data is required. To validate the effectiveness of cost evaluation criteria and the candidate set, the convincing discussion or experiments are required.

**Strengths And Weaknesses:**

strengths:
The active learning problem is important. The authors proposed to use cost as a new way to evaluate the importance of samples that would be annotated. It is novel. The motivation of candidate set is to improve the performance of annotator.

Weakness:
My primary concern lies in the relationship between the estimated cost and the actual number of labeled samples. After all, the goal of active learning is to minimize the number of labeled samples needed to achieve a model with the desired performance.

Also, it remains unclear how the performance of the trained model is guaranteed. The 1-\alpha^* probability only ensures that the annotator is provided with a candidate set containing the ground-truth class.

Here is also a related paper about active learning which authors may consider including in the related works. Uncertainty-Based Active Learning for Reading Comprehension. Wang, Jing and Shen, Jie and Ma, Xiaofei and Arnold, Andrew. TMLR, 2022.

---

> ### Author Response · Authors · 2025-06-25
> **Response to Reviewer jH6f (Part 1/2)**
>
> ### **General reply**
> We appreciate the reviewer's insightful feedback. Please find our responses to your concerns and comments below. Should any questions remain, we welcome further inquiry and will gladly provide additional elaboration.
>
> ---
>
> ### **Weakness 1. Clarifying the relationship between labeling cost and sample count**
> > My primary concern lies in the relationship between the estimated cost and the actual number of labeled samples. After all, the goal of active learning is to minimize the number of labeled samples needed to achieve a model with the desired performance.
>
> Thank you for your comment.
>
> Following the cost-sensitive active learning literature (Herde et al., 2021; Hu et al., 2020; Settles et al., 2008), we shift the objective of active learning from minimizing the number of labeled samples to minimizing annotation time, which more directly reflects financial and labor costs. Accordingly, we model annotation time from an information-theoretic perspective (Hu et al., 2020), and formulate our active learning scenario to minimize this cost. Our evaluation protocol is grounded in this cost measure, and we empirically validate the cost model through a user study (Table 1).
>
> CSQ reduces the annotation cost by shrinking the label search space presented to annotators. We emphasize that the effect of CSQ is orthogonal to the sampling strategy, meaning that regardless of how samples are selected, CSQ further reduces the annotation cost. Under a fixed annotation budget, this enables more samples to be labeled compared to conventional query (CQ), thereby improving overall cost-efficiency (Figure 2).
>
> We further clarify that the estimated cost refers to the cost per data point used by the proposed sampling strategy to favor low-cost samples. Thus, there is no direct correlation between the estimated cost and the number of labeled samples. However, we empirically observed that under a fixed cost budget, labeling multiple low-cost samples can be more beneficial for model training than labeling a single high-cost one. This principle is reflected in our sampling strategy and supported by the results shown in Table 2.
>
> We will clarify this assumption and motivation more explicitly in the revised manuscript.
>
> **References**
>
> - Hengtong Hu, Lingxi Xie, Zewei Du, Richang Hong, and Qi Tian. One-bit supervision for image classification. NeurIPS, 2020
>
> - Herde, M., Huseljic, D., Sick, B., & Calma, A. (2021). A Survey on Cost Types, Interaction Schemes, and Annotator Performance Models in Selection Algorithms for Active Learning in Classification. IEEE Access, 9, 166970–166989.
>
> - Settles, B., Craven, M., & Friedland, L. (2008). Active Learning with Real Annotation Costs. In NIPS Workshop on Cost-Sensitive Learning.
>
> ---
>
> ### **Weakness 2. Performance guarantee of the trained model**
> > It remains unclear how the performance of the trained model is guaranteed. The 1-\alpha^* probability only ensures that the annotator is provided with a candidate set containing the ground-truth class.
>
> While the $1−\alpha^*$ coverage ensures that the candidate set is likely to include the ground-truth label, it does not in itself guarantee the performance of the trained model. Instead, as shown in Theorem 3.1, it provides a condition under which the labeling cost is reduced by the candidate set query (CSQ).
>
> Unfortunately, our acquisition function lacks a theoretical guarantee for model performance at this point since it relies on heuristic informativeness measures such as entropy and BADGE (Ash et al., 2020), and due to the inherent complexity of deep learning models. For the same reason, to the best of our knowledge, recent deep active learning methods do not guarantee performance theoretically and focus primarily on empirical validation (Ash et al., 2020; Yehuda et al., 2022; Kim et al., 2023).
>
> To avoid potential misunderstanding, we clarify that CSQ does not affect the supervision used during training and is therefore not directly related to guaranteeing model performance. Regardless of whether the candidate set includes the ground-truth class, the annotator provides a single correct label, resulting in full supervision for every queried sample, as described in Section 3.1.
>
> **References**
>
> - Ash, J. T., Zhang, C., Krishnamurthy, A., Langford, J., & Agarwal, A. (2020). Deep Batch Active Learning by Diverse, Uncertain Gradient Lower Bounds. International Conference on Learning Representations.
> - Yehuda, O., Dekel, A., Hacohen, G., & Weinshall, D. (2022). Active learning through a covering lens. Advances in Neural Information Processing Systems, 35, 22354–22367.
> - Yoon-Yeong Kim, Youngjae Cho, JoonHo Jang, Byeonghu Na, Yeongmin Kim, Kyungwoo Song, Wanmo Kang, and Il-Chul Moon. Saal: sharpness-aware active learning. In Proc. International Conference on Machine Learning (ICML), 2023.

---

> ### Author Response · Authors · 2025-06-25
> **Response to Reviewer jH6f (Part 2/2)**
>
> ### **Weakness 3. Related work suggestion**
> > A related paper about active learning which authors may consider including in the related works. Uncertainty-Based Active Learning for Reading Comprehension. Wang, Jing and Shen, Jie and Ma, Xiaofei and Arnold, Andrew. TMLR, 2022.
>
> Thank you for letting us know about the related work. We will include and discuss it in the related work section of the revised manuscript!

---

> > ### Comment · Reviewer_jH6f · 2025-06-27
> > **technical question on main theorem**
> >
> > Thank you for addressing my prior questions. When checking the main theorem, Theorem 3.1, I found a technical issue that needs authors to clarify.
> >
> > Suppose that given x, Y(x) is obtained by randomly picking k class labels out of L. Then it is easy to show that
> >
> > alpha = 1 - k/L.
> >
> > Thus, the condition (2) reads as
> >
> > log(k+1)/log(L) < k/L,
> >
> > or equivalently
> >
> > log(k+1)/k < log(L)/L.
> >
> > But this appears not holding since the function f(x) := log(x) / x is decreasing.

---

> > > ### Author Response · Authors · 2025-06-28
> > > **Follow-up Response to Reviewer jH6f**
> > >
> > > To summarize your concern, you have pointed out a scenario in which Eq. (2) may not hold. We would like to clarify that Eq. (2) is **a sufficient condition** for reducing labeling cost, rather than a universal statement meant to apply to any arbitrary values of $k$, $L$, and $\alpha$. Specifically, Eq. (2) illustrates the condition under which the candidate set query (CSQ) is expected to reduce labeling costs relative to the conventional query (CQ). Consequently, it is natural and expected that Eq. (2) may not hold when the candidate set is randomly constructed.
> > >
> > > Furthermore, as detailed in Remark 3.2, we offer explicit and practical conditions under which CSQ achieves cost savings compared to CQ. For instance, with $L=1000$, a top-1 accuracy of at least $10.0$% ensures cost efficiency. Importantly, due to the use of conformal prediction, CSQ in practice reduces the labeling cost even under more relaxed conditions since it does not rely solely on top-1 predictions.
> > >
> > > In summary, Eq. (2) serves as a theoretical tool for verifying the conditions that enable CSQ to reduce labeling costs, and is not intended to be a universally applicable condition for all possible parameter combinations. We appreciate your thoughtful comment and will address this point more clearly in the revision.

---

> > > > ### Comment · Reviewer_jH6f · 2025-06-28
> > > >
> > > > The authors seem not getting the critical point. I reiterate below.
> > > >
> > > > Since f(x) = log(x) / x is decreasing in [0, +inf), and in your setup k < L, we always have
> > > >
> > > > log(k) / k > log(L) / L
> > > >
> > > > Thus the condition (2) will never be satisfied, making the entire theoretical result, Theorem 3.1, collapse.
> > > >
> > > > As this theorem is incorrect, you need to remove it or save it with new sufficient conditions.

---

> > > > > ### Author Response · Authors · 2025-06-29
> > > > > **Follow-up Response to Reviewer jH6f**
> > > > >
> > > > > Thank you again for your thoughtful and detailed follow-up. We recognized that your point is valid: Given the assumption that $k$ classes are randomly sampled from a uniform distribution over $L$ classes, i.e., $\alpha = 1 - k/L$, we agree with you that the inequality in Eq. (2) will not hold for any $k \leq L$, due to the monotonicity of $\frac{\log(x)}{x}$.
> > > > >
> > > > > However, **this does not mean that the theorem is incorrect** since (1) your assumption does not hold always, and more importantly, (2) even in the case where your assumption holds and thus the inequality does not hold, the theorem remains correct as the inequality is the sufficient condition in the theorem.
> > > > >
> > > > > (1) Your assumption does not hold always.
> > > > >
> > > > > You assumed that $k$ classes are randomly sampled from a uniform distribution over $L$ classes. However, this assumption often does not hold in active learning scenarios since active learning methods including ours exploit a model trained for the target task using a (even tiny) part of training data, which is highly likely to include the true class within its top-$k$ predictions where $k << L$. (Random sampling from the uniform distribution is not a reasonable approach at all to establishing a candidate class set in this regard.) In Remark 3.2, we offer explicit and practical conditions under which CSQ achieves cost savings compared to CQ. For instance, with $L=1000$, a top-1 accuracy of at least $10.0$% ensures cost efficiency, which is quite a mild condition. Importantly, due to the use of conformal prediction, CSQ in practice reduces the labeling cost even under more relaxed conditions since it does not rely solely on top-1 predictions.
> > > > >
> > > > > (2) Even in the case where your assumption holds and thus the inequality does not hold, the theorem remains correct.
> > > > >
> > > > > We would like to emphasize once again that the inequality in Eq. (2) is **the sufficient condition** for $C_\textrm{csq} (L, x, \alpha) < C_\textrm{con} (L, x)$. The main statement of Theorem 3 is: If $\frac{\log_2 (k+1)}{\log_2 L} < 1 - \alpha$, then $C_\textrm{csq} (L, x, \alpha) < C_\textrm{con} (L, x)$. This is a conditional statement with the form of “if $A$ then $B$” (or "$A \rightarrow B$" in logical expression), which means that $B$ must be true whenever $A$ is true. Hence, such a conditional statement is true when the hypothesis (i.e., sufficient condition) $A$ does not hold, regardless of the truth value of the consequence $B$, since it does not offer any guarantee regarding $B$ in this case.
> > > > >
> > > > > Accordingly, Theorem 3 of our paper states that if the inequality in Eq. (2) holds, CSQ will reduce the labeling cost, but if the inequality is not satisfied, it offers no guarantee regarding the labeling cost reduction. In other words, the theorem does not guarantee the cost reduction when candidate sets are constructed at random (as in your assumption) as the model has no predictive power. Hence, even in the case where the inequality does not hold, the theorem remains correct.

---

> > > > > > ### Author Response · Authors · 2025-06-29
> > > > > > **Follow-up Response to Reviewer jH6f**
> > > > > >
> > > > > > Dear Reviewer jH6f,
> > > > > >
> > > > > > We appreciate your dedicated service and detailed comments. We have noticed our previous reply to your final comment was not well-organized, so we have rewritten it below. We apologize for the trouble and would be grateful if you could check this response instead.
> > > > > >
> > > > > > ---
> > > > > >
> > > > > > Thank you again for your thoughtful and detailed follow-up. We recognized that your point is valid: given the assumption that $k$ classes are randomly sampled from a uniform distribution over $L$ classes, i.e., $\alpha = 1 - k/L$, we agree with you that the inequality in Eq. (2) will not hold for any $k \leq L$, due to the monotonicity of $\frac{\log(x)}{x}$.
> > > > > >
> > > > > > However, **this does not mean that the theorem is incorrect**. Even in the case where your assumption holds and thus the inequality does not hold, the theorem remains correct since the inequality is **the sufficient condition** for the conclusion $C_\textrm{csq} (L, x, \alpha) < C_\textrm{con} (L, x)$.
> > > > > >
> > > > > > The main statement of Theorem 3 is: If $\frac{\log_2 (k+1)}{\log_2 L} < 1 - \alpha$, then $C_\textrm{csq} (L, x, \alpha) < C_\textrm{con} (L, x)$. This is a conditional statement with the form of “if $A$ then $B$” (or "$A \rightarrow B$" in logical expression), which means that $B$ must be true whenever $A$ is true. In other words, such a conditional statement is true when the hypothesis (i.e., sufficient condition) $A$ does not hold, regardless of the truth value of the consequence $B$, since it does not offer any guarantee regarding $B$ in this case.
> > > > > >
> > > > > > Accordingly, Theorem 3 of our paper states that if the inequality in Eq. (2) holds, CSQ will reduce the labeling cost, but if the inequality is not satisfied, it offers no guarantee regarding the labeling cost reduction. In other words, the theorem does not discuss the cost reduction when candidate sets are constructed at random (as in your assumption) as the model has no predictive power. Hence, even in the case where the inequality does not hold, the theorem remains correct.
> > > > > >
> > > > > > So Theorem 3 is mathematically sound, then one may raise the question of whether its sufficient condition $\frac{\log_2 (k+1)}{\log_2 L} < 1 - \alpha$ is met in real active learning settings, thus guaranteeing reduced expected labeling costs in practice. To answer this question, in Remark 3.2, we offer explicit and practical conditions under which CSQ achieves cost savings compared to CQ. For instance, with $L=1000$, a top-1 accuracy of at least $10.0$% ensures cost efficiency, which is quite a mild condition. Moreover, due to the use of conformal prediction, CSQ in practice reduces the labeling cost even under more relaxed conditions since it does not rely solely on top-1 predictions.
> > > > > >
> > > > > > Again, we admit that the sufficient condition never holds under your assumption that $k$ classes are randomly sampled from a uniform distribution over $L$ classes. However, we would like to emphasize that this assumption is not likely in general active learning settings. Active learning methods including ours exploit a model trained for the target task using a (even tiny) part of training data, whose top-$k$ prediction highly likely to include the true class where $k << L$; random sampling from the uniform distribution is not at all a reasonable approach to establishing a candidate class set in this regard.

---

> > > > > > ### Comment · Reviewer_jH6f · 2025-06-29
> > > > > >
> > > > > > I strongly suggest authors consult someone with basic math background before getting into the discussion loop.
> > > > > >
> > > > > > What you were doing is something like
> > > > > >
> > > > > > "If x > x + 1, then 0 > 1 -- we have proved a revolutionary theorem based on a sufficient condition!"

---

> > > > > > > ### Author Response · Authors · 2025-06-30
> > > > > > > **Follow-up Response to Reviewer jH6f**
> > > > > > >
> > > > > > > TL;DR -- The inequality in Eq. (2) does not hold given a specific and very unlikely assumption, and it does not mean the inequality never holds.
> > > > > > >
> > > > > > > ---
> > > > > > >
> > > > > > > At the beginning of the first comment on Theorem 3.1, you stated:
> > > > > > > >Suppose that given $x$, $Y(x)$ is obtained by randomly picking $k$ class labels out of $L$.
> > > > > > >
> > > > > > > In other words, **you first assumed that $k$ classes are randomly sampled from a uniform distribution over $L$ classes**, to show that the inequality in Eq. (2) does not hold. Even if your conclusion based on this assumption is valid, it merely implies that the inequality in Eq. (2) does not hold under specific (and very unlikely) circumstances; **it does not mean the inequality never holds**. In fact, one can very easily find conditions where the inequality *does* hold as described below.
> > > > > > >
> > > > > > > The right-hand side of the inequality, i.e., $1-\alpha$, is the top-$k$ accuracy of the model since $\alpha$ is the probability that the candidate set of size $k$ does not include the true label. Suppose the target problem is 1,000-class classification ($L=1000$) and we consider top-10 predictions of a model ($k$ is fixed to 10) for CSQ. Then, the inequality holds when the top-10 accuracy of the model is about 34.7\% ($\frac{\log_2 (10+1)}{\log_2 1000}$), which is quite a mild condition. Remark 3.2 of the paper provides more of such conditions that hold in practice, as already stated in the previous response.
> > > > > > >
> > > > > > > We would again emphasize that your assumption of random sampling is very unlikely in general active learning settings since it implies that the model has no predictive power. Please note that the basic idea behind active learning is to leverage the model's predictive power to minimize labeling costs.
> > > > > > >
> > > > > > > >What you were doing is something like "If x > x + 1, then 0 > 1 -- we have proved a revolutionary theorem based on a sufficient condition!"
> > > > > > >
> > > > > > > This is not true at all. It rather sounds as if you are arguing that the proposition
> > > > > > > "If $x > 0$ , then $x/2 > 0$, for $x\in \mathbb{R}$" is invalid since the sufficient condition $x > 0$ never holds when $x$ is a negative number in $\mathbb{R}$.

---

> > > > > > > ### Comment · Editors_In_Chief · 2025-07-22
> > > > > > >
> > > > > > > Dear reviewer,
> > > > > > >
> > > > > > > We really appreciate your extensive engagement with the authors on this paper!
> > > > > > >
> > > > > > > "I strongly suggest authors consult someone with basic math background before getting into the discussion loop": While disagreements are often a part of academic discourse and can sometimes be frustrating, we kindly request that reviews and discussions focus on the content rather than the authors.
> > > > > > >
> > > > > > > Regards,
> > > > > > > EiCs

---

> > > ### Author Response · Authors · 2025-07-01
> > > **Summary of the Discussion on Theorem 3.1**
> > >
> > > We are providing a summary of discussions because we believe it will be helpful for the AE and other reviewers, as well as Reviewer jH6f.
> > >
> > > ---
> > >
> > > Theorem 3.1 is simply expressed as $A \rightarrow B$ given the following propositions:
> > > - $A:=\frac{\log_2 (k+1)}{\log_2 L}<1-\alpha$
> > > - $B:=C_\textrm{csq}(L,x,\alpha) < C_\textrm{con}(L,x)$
> > >
> > > The objection of Reviewer jH6f is that $A$ does not hold when $1-\alpha= k/L$. This condition, $1-\alpha= k/L$, is derived from the assumption that $k$ classes are *randomly sampled* from a uniform distribution over $L$ classes, as mentioned in his/her comment entitled "technical question on main theorem." Here is the statement quoted in the comment.
> > > >Suppose that given $x$, $Y(x)$ is obtained by randomly picking $k$ class labels out of $L$.
> > >
> > > We have addressed this misunderstanding as follows:
> > > 1. We have shown that **the proposition $A$ does hold under realistic, and quite mild conditions**. These conditions are immediately derived by replacing $k$ and $L$ in Eq. (2) with specific values from real experimental settings. More importantly, such conditions were already presented in Remark 3.2 of the paper.
> > > 2. We have explained that **the assumption made by the reviewer is very unlikely in general active learning scenarios**. The random sampling implies that the model being trained lacks predictive power, which contradicts the fundamental principle of active learning that aims to train the model within the annotation budget by leveraging the model's predictive power.
> > > 3. Furthermore, we pointed out that **the assumption for the main objection stems from a misunderstanding**. In the comment titled "the inequation is wrong," the reviewer explains his/her reasoning for the assumption as follows (quoted in the list): "There is no assumption in Theorem 3.1 about *selecting one out of $L$ possible options*, the default is random selection." But the phrase "selecting one out of $L$ possible options" is a part of the definition of the cost model we adopted, and it is completely unrelated to the proposition $A$ and to what the theorem asserts.
> > > 4. **Even given that unlikely assumption, the theorem itself still holds** since the proposition $A$ is the sufficient condition. In other words, the theorem does not discuss the cost reduction (the proposition $B$) when candidate sets are constructed at random (as in the assumption of the reviewer).

---

### Review · Reviewer_FogS · 2025-06-14

**Summary Of Contributions:**

The paper deals with **pool-based active learning**, where the objective is to maximize the performance of a machine learning model, while minimizing the labeling costs for collecting the associated training dataset. Therefore, existing active learning strategies aim to select unlabeled samples whose labeling and inclusion into the labeled training set result in the largest performance gains for the machine learning model. Typically, these strategies assume that a selected sample is presented to a labeler whose task is to decide on the ground truth class from the list of candidate classes. This decision often gets more time-intensive (and therefore costly), when there are many candidate classes. The main **contributions** of the paper to resolve this issue are the following ones:

- Candidate set query as an alternative query type is proposed, which leverages conformal predictions of the machine learning model to propose only small subsets of candidate classes per sample.
- A well-motivated labeling cost model accounts for the size of each subset of candidate classes and lays the foundation for a new cost-sensitive active learning framework, which can be combined with existing active learning strategies providing sample-wise scores.
- Extensive experimental results across varying settings, e.g., standard image and text classification, imbalanced classification, and classification with uniform label noise, demonstrate the benefit of this framework for cost-sensitive active learning.

**Audience:**

Yes

**Broader Impact Concerns:**

Humans were involved in a case study for labeling. However, the number of images to be labeled seemed small and their content probably unproblematic.

**Claims And Evidence:**

Yes

**Requested Changes:**

I would appreciate it if the authors could address the abovementioned weaknesses and resolve the following minor remarks.

**Minor Remarks:**
- Resolve Inconsistent capitalization of "Initial" in "Require: ..." of Algorithm 1.
- Update arXiv references of LabelBench to its formal publication at DMLR.
- Check consistent venue referencing, e.g., inconsistent usage of "(NeurIPS)".

**Strengths And Weaknesses:**

Below, I list the strengths and weaknesses according to their importance, starting with the most important one and ending with the least important one. Thereby, the strengths are rather concisely formulated, serving primarily as confirmation and, unlike the weaknesses, requiring no further discussion. Of course, I will reconsider my perspective of my comments, if I have misunderstood or overlooked any aspect. Looking forward to a constructive discussion!

**Strengths:**
- The paper addresses a well-known practical problem [1, 2] of designing more efficient queries for active learning.
- Leveraging conformal predictions for creating candidate subsets of classes is a neat idea to balance the trade-off between small subsets and the high probability of keeping the ground truth class as candidate.
- The proposed cost model is supported by theoretical insights and a case study with humans as labelers.
- The experimental evaluation is well designed, very extensive and shows in combination with a detailed ablation study the benefits of the cost-sensitive active learning framework's components.
- The authors provide code including detailed descriptions improving the reproducibility.
- The writing and presentation of the paper is very clear.

**Weaknesses:**
- Although ResNets remain popular, they’re now fairly dated. At the same time, active learning studies are increasingly built on features from pretrained (often self‐supervised) models [3, 4]. ProbCover even adopts this approach, as noted in your code as part of the supplementary material. That raises a natural question: why not use these modern pretrained networks as backbones, either in combination with fine-tuning or linear probing for evaluation? The same logic applies to case study with the text dataset where you used an SVM. Here, large language models could serve as powerful feature extractors. In fact, incorporating such state-of-the-art models typically means you can work with much smaller per-round budgets, since they need far fewer labeled examples to reach convergence.
- The case study with real participants is a valuable contribution. However, the case study’s protocol in the appendix (see quote) of always including the ground-truth label in each candidate set leaves the investigation open whether omitting the correct label from the sets of candidate classes could irritate them or even bias their decisions.
> "For small-sized candidate sets, we ensure the inclusion of the ground truth by randomly trimming around it when generating the candidate sets."
- Your hyperparameter $d$ controlling the influence of the acquisition function is set to $d=0.3$ for all experiments with image data. At the same time this is the smallest and seems to be the best value according to your ablation study in Figure 11. Therefore, it would be interesting to see what happens for an even smaller value. Moreover, I am confused why and how did you choose $d=1.2$ for the experiments with the text dataset?
- Your cost model relies only on the subset's size for the class candidates. Your case study confirms that this is a good proxy. Moreover, your conformal predictions automatically penalize too uncertain samples because they will have larger subsets. However, I think a brief discussion (e.g., as part of the related work) regarding the existing literature on more complex cost models for the annotation time would be more transparent regarding potential limitations. For example, there are several resources [2, 5, 6, 7] demonstrating/overviewing the non-negligible effects of sample-wise labeling difficulties and the labeler's behavior.

**References:**
- [1] B. Settles, ‘‘From theories to queries: Active learning in practice,’’ in Proc. Workshop Active Learn. Exp. Design, Sardinia, Italy, 2011, pp. 1–18
- [2] Herde, M., Huseljic, D., Sick, B., & Calma, A. (2021). A Survey on Cost Types, Interaction Schemes, and Annotator Performance Models in selection algorithms for active learning in classification. IEEE Access, 9, 166970-166989.
- [3] Hacohen, G., Dekel, A., & Weinshall, D. (2022, June). Active Learning on a Budget: Opposite Strategies Suit High and Low Budgets. In International Conference on Machine Learning (pp. 8175-8195). PMLR.
- [4] Gupte, S. R., Aklilu, J., Nirschl, J. J., & Yeung-Levy, S. (2024) Revisiting Active Learning in the Era of Vision Foundation Models. Transactions on Machine Learning Research.
- [5] B. Settles, M. Craven, and L. Friedland, ‘‘Active learning with real annotation costs,’’ in Proc. NIPS Workshop Cost-Sensitive Learn., Vancouver, CA, Canada, 2008, pp. 1–10.
- [6] B. C. Wallace, K. Small, C. E. Brodley, and T. A. Trikalinos, ‘‘Who should label what? Instance allocation in multiple expert active learning,’’ in Proc. SIAM Int. Conf. Data Mining, Apr. 2011, pp. 176–187.
- [7] Arora, S., Nyberg, E., & Rose, C. (2009). Estimating annotation cost for active learning in a multi-annotator environment. In Proceedings of the NAACL HLT 2009 Workshop on Active Learning for Natural Language Processing (pp. 18-26).

---

> ### Author Response · Authors · 2025-06-25
> **Response to Reviewer FogS (Part 1/2)**
>
> ## **General reply**
>
> We sincerely appreciate your detailed and insightful feedback. Below, we provide point-by-point responses to your comments. If any concerns remain, we would be happy to offer further clarification.
>
> ---
>
> ### **Weakness 1. CSQ with advanced architectures and self-supervised pretraining**
> > Although ResNets remain popular, they’re now fairly dated. At the same time, active learning studies are increasingly built on features from pretrained (often self‐supervised) models [3, 4]. ProbCover even adopts this approach, as noted in your code as part of the supplementary material. That raises a natural question: why not use these modern pretrained networks as backbones, either in combination with fine-tuning or linear probing for evaluation? The same logic applies to case study with the text dataset where you used an SVM. Here, large language models could serve as powerful feature extractors. In fact, incorporating such state-of-the-art models typically means you can work with much smaller per-round budgets, since they need far fewer labeled examples to reach convergence.
>
> Thank you for your comment.
>
> We agree that CSQ can benefit from advanced architectures and self-supervised pretraining. To validate this, we evaluate CSQ with ResNet50 and ViT-S/16 pretrained via self-supervised learning (Caron et al., 2021) on ImageNet-1K. As shown in Figure 19 in the revised manuscript, CSQ consistently outperforms baselines under these settings and shows strong performance even in early rounds. This confirms that CSQ effectively leverages modern backbones and pretrained features.
>
> We also evaluate CSQ on text classification using RoBERTa as a frozen feature extractor. As shown in Figure 20 in the revised manuscript, CSQ reduces relative labeling cost by up to 5%p compared to the CQ baseline on the R52 dataset, further demonstrating its compatibility with modern pretrained language models.
>
>
> **References**
> - Caron, Mathilde, Hugo Touvron, Ishan Misra, Hervé Jégou, Julien Mairal, Piotr Bojanowski, and Armand Joulin. Emerging properties in self-supervised vision transformers. ICCV, 2021.
>
> ---
>
> ### **Weakness 2. User study with incorrect candidate sets**
> > The case study with real participants is a valuable contribution. However, the case study’s protocol in the appendix (see quote) of always including the ground-truth label in each candidate set leaves the investigation open whether omitting the correct label from the sets of candidate classes could irritate them or even bias their decisions.
>
> Thank you for the valuable suggestion.
>
> To investigate scenarios where the correct label is omitted from the candidate set, we have conducted an additional user study and are currently awaiting its results. We expect to collect the results shortly and will report the findings promptly.
>
> ---
>
> ### **Weakness 3. Effect of smaller hyperparameter $d$ values**
> > Your hyperparameter d controlling the influence of the acquisition function is set to d=0.3 for all experiments with image data. At the same time, this is the smallest and seems to be the best value according to your ablation study in Figure 11. Therefore, it would be interesting to see what happens for an even smaller value. Moreover, I am confused why and how did you choose d=1.2 for the experiments with the text dataset?
>
> Thank you for your comment.
>
> To further analyze the effect of the hyperparameter $d$ in Eq. (8), we conducted additional ablation studies using smaller $d$ values for both entropy and BADGE sampling on CIFAR-100. As shown in Figure 22 in the revised manuscript, performance improves with smaller d in entropy sampling, whereas BADGE shows saturation or even performance drops.
>
> In Figure 23 in the revised manuscript, we also analyze the effect of $d$ values on the text dataset. The best performance is observed at $d=2.0$, and no clear trend is seen as $d$ increases or decreases. The initial choice of $d = 1.2$ was made without tuning, loosely guided by earlier image-domain experiments. The new results confirm that performance is relatively insensitive to $d$, likely due to the use of fixed features in SVMs.
>
> Overall, the optimal $d$ varies by domain and acquisition strategy: $d = 0.3$ is robust for image tasks, while higher values such as $d = 2.0$ may slightly improve performance in the language domain. These findings confirm that our method remains effective across a reasonable range of $d$ values.

---

> ### Author Response · Authors · 2025-06-25
> **Response to Reviewer FogS (Part 2/2)**
>
> ### **Weakness 4. Related work on cost modeling**
> > Your cost model relies only on the subset's size for the class candidates. Your case study confirms that this is a good proxy. Moreover, your conformal predictions automatically penalize too uncertain samples because they will have larger subsets. However, I think a brief discussion (e.g., as part of the related work) regarding the existing literature on more complex cost models for the annotation time would be more transparent regarding potential limitations. For example, there are several resources [2, 5, 6, 7] demonstrating/overviewing the non-negligible effects of sample-wise labeling difficulties and the labeler's behavior.
>
> We thank the reviewer for the insightful and constructive comment. Based on the suggested references, we will include the following discussion in the related work section to clarify the scope and limitations of our current cost model:
>
> > **Realistic cost model for AL.** While we follow Hu et al. (2020) in modeling annotation cost based on the size of the labeling options using an information-theoretic perspective, other works have explored additional factors influencing real-world annotation cost, such as annotator behavior and interaction complexity (Settles et al., 2008; Arora et al., 2009; Wallace et al., 2011; Herde et al., 2021). For example, Settles et al. (2008) and Arora et al. (2009) show that instance, task, and user features jointly affect annotation time, while Wallace et al. (2011) propose a cost-saving strategy where less experienced annotators can pass uncertain cases to more experienced ones. Herde et al. (2021) provide a taxonomy of cost-aware active learning approaches under more realistic assumptions. These studies highlight that annotation cost depends on multiple factors and varies across settings. While our size-based proxy is effective and supported by user study, richer cost models may better capture real-world annotation behavior and further improve performance.
>
> **References**
>
> - Hengtong Hu, Lingxi Xie, Zewei Du, Richang Hong, and Qi Tian. One-bit supervision for image classification. NeurIPS, 2020
>
> - Herde, M., Huseljic, D., Sick, B., & Calma, A. (2021). A Survey on Cost Types, Interaction Schemes, and Annotator Performance Models in Selection Algorithms for Active Learning in Classification. IEEE Access, 9, 166970–166989.
>
> - Settles, B., Craven, M., & Friedland, L. (2008). Active Learning with Real Annotation Costs. In NIPS Workshop on Cost-Sensitive Learning.
>
> - Arora, S., Nyberg, E., & Rose, C. (2009). Estimating Annotation Cost for Active Learning in a Multi-Annotator Environment. In NAACL HLT Workshop on Active Learning for NLP, pp. 18–26.
> Wallace, B. C., Small, K., Brodley, C. E., & Trikalinos, T. A. (2011).
>
> - Who Should Label What? Instance Allocation in Multiple-Expert Active Learning. In Proc. SIAM Int. Conf. on Data Mining, pp. 176–187.
>
> ---
>
> ### **Requested Changes**
>
> Thank you for pointing out the issues regarding capitalization in Algorithm 1, citation formatting, and venue reference consistency. These have all been addressed in the revised manuscript.

---

> ### Author Response · Authors · 2025-06-27
> **Follow-up to Reviewer FogS**
>
> ### **Weakness 2. User study with incorrect candidate sets**
> > The case study with real participants is a valuable contribution. However, the case study’s protocol in the appendix (see quote) of always including the ground-truth label in each candidate set leaves the investigation open whether omitting the correct label from the sets of candidate classes could irritate them or even bias their decisions.
>
> As a follow-up to our earlier response, we conducted an additional user study with 14 participants investigating cases where the ground-truth label is omitted from the candidate set. Specifically, the questionnaire consists of 20 problems of choosing one out of 32 class labels, each with a distinct image and class label. To simulate the case where the correct label is omitted from the candidate set, we randomly select 10 problems and exclude the ground-truth label from the candidate set, resulting in a 50% inclusion rate. In such cases, participants were instructed to select the “None of the above” option, after which they were shown the remaining list of classes to choose the correct one. We set the inclusion rate to 50% to prevent participants from forming a bias about the presence of the correct label in the candidate set. As a baseline, we used the same 20 problems but presented all 32 class options in a conventional query format.
>
> Table R1 summarizes the results of this user study. CSQ improved the annotation accuracy when the ground-truth label was included in the candidate set since the narrowed candidate set helps annotators identify the correct label more reliably. Moreover, the accuracy remained comparable to that of the baseline even when the label was omitted from the candidate set since the two-step interface of CSQ ultimately presents the full set of classes, as in the baseline. We do not observe any notable labeling bias caused by omitting the correct label in the candidate set. Since candidate sets tend to include the ground-truth label under reasonably accurate model predictions (Figure 5c), CSQ is expected to reduce annotation errors in practice.
>
> #### **Table R1**: User-study results on a fixed set of 20 problems of choosing one out of 32 class labels. For each query type, 10 problems are configured such that the ground-truth label is included in the candidate set (**included**), and the other 10 omit it (**omitted**). While the candidate set query (CSQ) uses candidate sets of size $k$ selected from the 32 classes, the conventional query (CQ) does not rely on candidate sets. For a fair comparison, we apply the same included/omitted grouping of problems across all query types.
>
> | Query type                  | **Included** sample accuracy (\%) | **Omitted** sample accuracy (\%) |
> |:-----------------------------------------------|:------------------------------------:|:-------------------------------:|
> | Conventional query (CQ)                  |                  92.7 ± 7.9           |              97.5 ± 4.1        |
> | Candidate set query (CSQ): $k=4$   |            **100.0 ± 0.0**        |          **98.6 ± 4.9**      |
> | Candidate set query (CSQ): $k=8$   |                 97.1 ± 3.8          |        **98.6  ± 3.8**        |

---

### Decision · Action_Editor_bwpR · 2025-07-22

**Recommendation:** Accept as is

**Additional Comments:**

This paper introduces Candidate Set Query (CSQ) as an efficient method for active learning.
In CSQ, the algorithm presents k candidate class labels to an annotator.
The annotator then selects the appropriate label from this set.
If none of the k labels are suitable, the annotator choose from the remaining class labels.
The core idea is that if the initially presented k labels have a sufficiently high probability of containing the correct label, the labeling cost for the annotator can be significantly reduced.

The authors demonstrated the effectiveness of CSQ through experiments on CIFAR-10, CIFAR-100, and ImageNet64x64 datasets.
The authors have incorporated reviewer feedback by adding a user study and some ablation studies.
These additional results further strengthen the claim of CSQ's effectiveness.

During the reviewer discussion phase, the validity of Theorem 3.1 became a point of discussion.
While it's unfortunate that the discussion did not remain constructive, as an Area Editor, I have thoroughly reviewed Theorem 3.1 and found it to be sound.

Some reviewers also raised concerns regarding the impact of annotation errors and the applicability to larger datasets.
The authors have addressed these concerns to a reasonable extent by including some ablation studies.

In summary, although there are several concerns raised by the reviewers, the authors have appropriately updated the paper, providing reasonable responses to these points.

**Audience:**

Yes

**Audience Explanation:**

Active learning is a major research topic in machine learning and is of interest to the TMLR readers.

**Claims And Evidence:**

Yes

**Claims Explanation:**

This paper introduces Candidate Set Query (CSQ) as an efficient method for active learning.
In CSQ, the algorithm presents k candidate class labels to an annotator.
The annotator then selects the appropriate label from this set.
If none of the k labels are suitable, the annotator choose from the remaining class labels.
The core idea is that if the initially presented k labels have a sufficiently high probability of containing the correct label, the labeling cost for the annotator can be significantly reduced.

The authors demonstrated the effectiveness of CSQ through experiments on CIFAR-10, CIFAR-100, and ImageNet64x64 datasets.
The authors have incorporated reviewer feedback by adding a user study and some ablation studies.
These additional results further strengthen the claim of CSQ's effectiveness.

Overall, the presented results provide sufficient evidence to support the efficacy of CSQ.